# Practical Kernel Selection for Kernel-based Conditional Independence Test

**Wenjie Wang**[1]   **Mingming Gong**[1,5]   **Biwei Huang**[2]   **James Bailey**[1]
**Bo Han**[3]   **Kun Zhang**[4,5]   **Feng Liu**[1]*

[1]The University of Melbourne   [2]University of California, San Diego   [3]Hong Kong Baptist University
[4]Carnegie Mellon University   [5]Mohamed bin Zayed University of Artificial Intelligence

wenjie.wang3@student.unimelb.edu.au   {mingming.gong, feng.liu1}@unimelb.edu.au

## Abstract

Conditional independence (CI) testing is a fundamental yet challenging task in modern statistics and machine learning. One pivotal class of methods for assessing conditional independence encompasses kernel-based approaches, known for assessing CI by detecting general conditional dependence without imposing strict assumptions on relationships or data distributions. As with any method utilizing kernels, selecting appropriate kernels is crucial for precise identification. However, it remains underexplored in kernel-based CI methods, where the kernels are often determined manually or heuristically. In this paper, we analyze and propose a kernel parameter selection approach for the kernel-based conditional independence test (KCI). The kernel parameters are selected based on the ratio of the statistic to the asymptotic variance, which approximates the test power for the given parameters at large sample sizes. The search procedure is grid-based, allowing for parallelization with manageable additional computation time. We theoretically demonstrate the consistency of the proposed criterion while explicitly accounting for model estimation bias, which is a distinctive challenge specific to CI testing task. Furthermore, we conduct extensive experiments on both synthetic and real-world datasets to empirically validate the effectiveness of our method.

## 1   Introduction

Conditional independence (CI) test is a cornerstone of statistics and machine learning. Let $X, Y$ and $Z$ be the random variables, then the conditional independence relationship between $X$ and $Y$ given $Z$, denoted by $X \perp\!\!\!\perp Y \mid Z$, indicates that knowing the values of $Z$, the knowledge of $X$ does not yield any extra information about $Y$. This conditional independence relationship enables the removal of redundant variables when constructing probabilistic models for a given variable set. Therefore, the utilization of CI has expanded across diverse domains, including causality [Spirtes et al., 2000, 1995, Pearl et al., 2000, Huang et al., 2020, Chi et al., 2024], fairness representation learning [Mehrabi et al., 2021, Han et al., 2025], feature selection [Fukumizu et al., 2009, Song et al., 2012].

Traditional CI testing methods either address the discrete case [Margaritis, 2005] or rely on simplifying assumptions to handle the continuous case [Lawrance, 1976, Linton and Gozalo, 1996]. These assumptions can be restrictive, and when violated or when data are limited, these methods often yield biased estimates and erroneous inferences, leading to unreliable conclusions. Daudin [1980] extended the concept of partial correlation to general scenarios involving nonlinear and non-Gaussian noise, redefining conditional independence as the zero correlation of any regression residual functions

---

*Corresponding to Feng Liu. The code is available at: https://github.com/wenjiewang3/PowerKCI

within constrained $L^2$ spaces. While this definition can identify general CI relationships, it requires considering all possible functions within these constrained $L^2$ spaces, which is infeasible.

To make it practical, Zhang et al. [2011] relaxed the function spaces to reproducing kernel Hilbert spaces (RKHS) using kernel methods, simplifying computation while preserving the ability to capture general CI relationships. In kernel methods, there exists a special class known as characteristic kernels, [Fukumizu et al., 2007] such as the Gaussian and Laplace kernels, which are capable of measuring distributional homogeneity [Gretton et al., 2012a, Song et al., 2009]. Building on this, Zhang et al. [2011] introduce the Kernel-based Conditional Independence (KCI) statistic, which replaces regression residuals with kernel-based analogues. They adopt the framework of conditional mean embeddings (CME) [Song et al., 2009, Grünewälder et al., 2012] to model nonlinear relationships, and define the test statistic as the Hilbert-Schmidt norm of the cross-covariance operator between residuals in RKHS, replacing the original cross-covariance defined in $L^2$ space [Gretton et al., 2005a]. This formulation enables KCI to directly assess whether $P_{XY|Z}P_Z = P_{X|Z}P_{Y|Z}P_Z$, without requiring explicit estimation of complex conditional marginals. Due to the reproducing property of kernels, a zero value of this statistic implies that all partial correlations between residual functions, represented in the RKHS induced by the chosen kernels, are also zero. Consequently, by employing characteristic kernels whose RKHS are dense in $L^2$ [Sriperumbudur et al., 2008], KCI is capable of capturing a broad class of conditional dependencies, extending beyond the linear relationships. However, as with all kernel-based methods, the performance of KCI is highly sensitive to the choice of kernels.

It is well known that the effectiveness of kernel-based methods critically depends on the choice of kernels across a wide range of tasks [Brockmann et al., 1993, Chapelle and Vapnik, 1999], particularly the choice of kernel parameters [Schölkopf et al., 2002]. A commonly used strategy is the median heuristic, which sets the kernel bandwidth to the median of pairwise distances between data points. While simple and widely adopted, this approach is often suboptimal for a given dataset [Ramdas et al., 2015, Garreau et al., 2017]. Consequently, data-driven selection of kernel parameters is crucial for maximizing the performance of kernel-based methods. Gretton et al. [2012b] propose a statistic-to-variance ratio criterion that leverages the properties of U-statistics to estimate test power under given kernel parameters, and use this to guide kernel selection. Variants of this criterion have been extended to tasks such as two-sample testing [Liu et al., 2021, Biggs et al., 2023] and unconditional independence testing [Albert et al., 2022], with some methods further incorporating deep kernels [Liu et al., 2020, Xu et al., 2024]. These approaches perform continuous optimization to learn near-optimal or even oracle kernel choices for the tasks above. However, such optimization-based strategies are not directly applicable to conditional independence testing, which requires additional consideration of regression-induced estimation bias—a distinguishing challenge unique to the CI test task. As a result, suitable kernel selection methods for CI testing remain largely underexplored.

**Contributions.** In this paper, we propose a kernel selection method to optimize the kernel parameters involved in the widely used KCI statistic. Our criterion remains grounded in the statistic-to-variance ratio framework, but explicitly accounts for the characteristics of the CI task, which require indirect consideration of regression residuals and the associated estimation bias. To this end, we first decompose the original KCI statistic to isolate the kernel component associated with the conditioning set, which was previously entangled within the residuals. This decomposition effectively reduces such estimation bias, thereby enabling more effective kernel selection in subsequent steps. We use the ratio of the statistic to the asymptotic variance as a criterion, which approximates the test power at large sample sizes. Then, the kernel parameters are selected based on the maximum ratio from a list of potential candidates using a grid search approach. Unlike existing continuous optimization approaches, this practical search strategy accounts for regression estimation bias and leverages the parallelizability of regression learning under different kernel parameters. Theoretically, we provide the first convergence result for this power-based criterion under the conditional independence testing setting, explicitly accounting for model estimation bias. Extensive experiments on both synthetic and real data demonstrate the efficacy of our method with manageable computation time.

## 2 Preliminaries

### 2.1 Related Works

**CI testing and its hardness.** A central difficulty in conditional independence (CI) testing is fundamental: without extra structure, a valid test cannot be uniformly powerful against all alternatives

[Shah and Peters, 2020]. This has led to two pragmatic lines. One imposes side information (e.g., Model-X–style knowledge of part of the conditional law) to stabilize null simulation or calibration [Candes et al., 2018, Berrett et al., 2020, Doran et al., 2014]. The other residualizes $X$ and/or $Y$ on $Z$ and then tests for remaining dependence; kernel-based CI tests fall in this camp, using RKHS embeddings to assess higher-order partial dependence without parametric assumptions [Fukumizu et al., 2007, Sun et al., 2007, Zhang et al., 2011, Huang et al., 2022]. In both cases, bias from estimating conditionals (or their surrogates) flows directly into Type-I error control and power—unlike in two-sample or unconditional independence tests—so reducing residualization error is both distinctive and pivotal in CI testing.

**Kernel selection.** Kernel tests are highly sensitive to hyperparameters (often more than the kernel family), yet practice commonly defaults to the median-distance bandwidth, which can be suboptimal [Schölkopf et al., 2002]. Outside CI, data-driven criteria optimize kernels by maximizing a statistic or statistic-to-variance ratio that proxies power, yielding strong results in two-sample, unconditional independence, and goodness-of-fit testing, including with deep kernels [Fukumizu et al., 2009, Gretton et al., 2012b, Liu et al., 2021, Biggs et al., 2023, Albert et al., 2022, Ren et al., 2024a,b, Liu et al., 2020, Xu et al., 2024]. CI, however, is qualitatively different: the target is conditional structure and the test statistic typically depends on regression residuals (e.g., KCI/GCM), introducing additional estimation bias/variance beyond finite-sample variability of a direct discrepancy [Zhang et al., 2011, Shah and Peters, 2020]. This distinction limits the direct transfer of existing selection rules and motivates CI-aware kernel selection that explicitly couples kernel choice with the residualization step—aimed at better Type-I control and power for kernel-based CI. For more related papers and discussion, please refer to Appendix A.

## 2.2 Kernel-based measures of conditional dependence

**CI definition and testing procedure.** Suppose there are three random variables $X$, $Y$ and $Z$ with observational points, and their joint distribution is absolutely continuous with respect to Lebesgue measure with density $P$. The problem of testing CI between $X$ and $Y$ given $Z$ can be written in the form of a hypothesis testing:

$$\mathrm{H_0} : X \perp\!\!\!\perp Y \mid Z \quad \text{vs.} \quad \mathrm{H_1} : X \not\!\perp\!\!\!\perp Y \mid Z.$$

CI testing typically involves the following steps: define a test statistic $T$ and a significance level $\alpha$ (typically set at $0.05$); compute the observed statistic $\hat{T}$; calculate the $p$-value under $\mathrm{H_0}$; and reject $\mathrm{H_0}$ if the $p$-value is less than or equal to $\alpha$. We evaluate the performance of a CI testing method using Type I error (False Positive) and Type II error (False Negative): a reliable CI test controls the Type I error below the significance level while minimizing the Type II error.

We provide the general characterization of conditional independence from the perspective of partial association:

**Definition 1.** [Daudin, 1980] Random variables $X$ and $Y$ are independent conditioned on $Z$, denoted $X \perp\!\!\!\perp Y \mid Z$, if for all functions $g \in L^2_{XZ}$ and $h \in L^2_Y$, we have almost surely in $Z$ that

$$\mathbb{E}[g(X,Z)\, h(Y) \mid Z] = \mathbb{E}[g(X,Z) \mid Z]\mathbb{E}[h(Y) \mid Z].$$

**Theorem 2.** [Daudin, 1980] $X \perp\!\!\!\perp Y \mid Z$ if and only if

$$\mathbb{E}[g(X,Z)h(Y)] = 0 \quad \forall g \in E_1, h \in E_2, \tag{1}$$

where $E_1 = \{g \in L^2_{XZ} : \mathbb{E}[g(X,Z) \mid Z] = 0\}$ and $E_2 = \{h \in L^2_Y : \mathbb{E}[h(Y) \mid Z] = 0\}$.

Since $g(X,Z)$ can represent any general relationship between $X$ and $Z$, Theorem 2 can be intuitively understood as asserting that the residuals obtained from regressing any function mappings of $(X,Z)$ and $Y$, defined in the $L^2$ space, onto $Z$ are uncorrelated. Therefore, this definition can capture general CI relationships but requires considering all possible functions in $L^2$.

**Kernel-based CI statistic (KCI).** To use this characterization in practice, Zhang et al. [2011] introduce it within the RKHS. For the random variable $X$ with its domain $\mathcal{X}$, we define the RKHS $\mathcal{H}_{\mathcal{X}}$ on $\mathcal{X}$ with a symmetric positive-definite function $k_{\mathcal{X}} : \mathcal{X} \times \mathcal{X} \to \mathbb{R}$. The kernel can be represented as an inner product in $\mathcal{H}_{\mathcal{X}}$ via a mapping $\phi_x : \mathcal{X} \to \mathcal{H}_{\mathcal{X}}$, which is $k_{\mathcal{X}}(x,x') = \langle \phi_x(x), \phi_x(x') \rangle$. And with the reproducing property, we have $\forall x \in \mathcal{X}$ and $\forall f \in \mathcal{H}_{\mathcal{X}}, f(x) = \langle f, \phi_x(x) \rangle$. Similar to the notation on $X$, we define $(k_{\mathcal{Y}}, \phi_y(Y), \mathcal{H}_{\mathcal{Y}}), (k_{\mathcal{Z}}, \phi_z(Z), \mathcal{H}_{\mathcal{Z}})$ and $(k_{\mathcal{XZ}}, \phi_{xz}(X,Z), \mathcal{H}_{\mathcal{XZ}})$ with

$k_{\mathcal{XZ}} \coloneqq k_{\mathcal{X}} k_{\mathcal{Z}}$. Building upon the cross-covariance operator [Fukumizu et al., 2007], Zhang et al. [2011] then propose the Kernel-based Conditional Independence (KCI) statistic for CI testing, which is defined as follows:

$$\Sigma_{\ddot{X}Y|Z} = \mathbb{E}[(\phi_{xz}(X,Z) - \mu_{XZ|Z}(Z)) \otimes (\phi_y(Y) - \mu_{Y|Z}(Z))], \qquad (2)$$

where $\ddot{X} \coloneqq (X,Z)$, $\otimes$ is the tensor product, $\mu_{XZ|Z}$ and $\mu_{Y|Z}$ represent the conditional mean embeddings given by $\mu_{XZ|Z}(Z) = \mathbb{E}[\phi_{xz}(X,Z) \mid Z]$ and $\mu_{Y|Z}(Z) = \mathbb{E}[\phi_y(Y) \mid Z]$. Utilizing the property that for any $g \in \mathcal{H}_{\mathcal{XZ}}$ and $h \in \mathcal{H}_{\mathcal{Y}}$ (see e.g. Gretton [2013, Lecture 5]), the tensor product operates as $(\phi_{xz} \otimes \phi_y)g = \langle \phi_{xz}, g \rangle \phi_y$, we can derive the following equation:

$$\left\langle h, \Sigma_{\ddot{X}Y|Z} g \right\rangle = \mathbb{E}[(g(X,Z) - \mathbb{E}[g(X,Z) \mid Z])(h(Y) - \mathbb{E}[h(Y) \mid Z])],$$

which holds for any $g \in \mathcal{H}_{\mathcal{XZ}}$ and $h \in \mathcal{H}_{\mathcal{Y}}$. For a class of kernel functions known as characteristic kernels (such as Gaussian kernel), their RKHSs are dense in $L^2$ spaces [Sriperumbudur et al., 2008]. With characteristic kernels employed, if $\Sigma_{\ddot{X}Y|Z} = 0$, Eq. 1 holds for any $g \in E_1 \cap \mathcal{H}_{\mathcal{XZ}}$ and $h \in E_2 \cap \mathcal{H}_{\mathcal{Y}}$, encompassing sufficient functions by continuity and density. This implies that $\Sigma_{\ddot{X}Y|Z} = 0$ if and only if $X \perp\!\!\!\perp Y \mid Z$. Therefore, we can test conditional independence by evaluating whether the Hilbert-Schmidt norm of the operator is zero, i.e. $\|\Sigma_{\ddot{X}Y|Z}\|_{\mathrm{HS}}^2 = 0$.

## 3 Power-based kernel selection for conditional independence testing

In all kernel-involved methods, the choice of kernel parameters is crucial, and KCI is no exception. The original KCI relies on the median heuristic to determine its kernel parameters, which does not fully capture the inherent characteristics of the data. In this section, we introduce our power-based kernel selection method for KCI, named *Power*.

**Decomposition of KCI.** We begin our method by decomposing the kernel mapping of the conditioning set $Z$ from the concatenated $\phi_{zx}(X,Z)$ in its original form (i.e., Eq. 1), as suggested by Pogodin et al. [2022, 2024]. According to [Mastouri et al., 2021], the RBF kernels (e.g. the Gaussian and Laplace kernel) of $\phi_{zx}(X,Z)$ can be decomposed into $\phi_x(X) \otimes \phi_z(Z)$. For the conditional expectation, we can derive that $\mu_{XZ|Z}(Z) = \mathbb{E}[\phi_x(X) \otimes \phi_z(Z) \mid Z] = \mathbb{E}[\phi_x(X) \mid Z] \otimes \phi_z(Z)$. Then, the decomposed form of the KCI statistic, which isolates $\phi_z(Z)$ from the regression residual of $\phi_{xz}(X,Z)$ with respect to $Z$, has the following form [Pogodin et al., 2024]:

$$\Sigma_{\ddot{X}Y|Z} = \mathbb{E}[(\phi_x(X) - \mu_{X|Z}(Z)) \otimes (\phi_y(Y) - \mu_{Y|Z}(Z)) \otimes \phi_z(Z)]. \qquad (3)$$

This decomposition significantly reduces the estimation error of the conditional mean embedding involved in KCI. Theoretically, it avoids estimating the identity operator $\mu_{Z|Z} = \phi(Z)$, which is not norm-bounded and, therefore, not a Hilbert-Schmidt operator, leading to an ill-specified regression problem. Empirically, under finite samples, if we approximate it as a finite-dimensional vector-valued regression, the estimation of $\mu_{XZ|Z}$ in its original form is much less smooth compared to $\mu_{X|Z}$. This implies a lower $\beta$ value (slower decay rate of the function's eigenvalues) in vector-valued regression [Fischer and Steinwart, 2020, Li et al., 2022], which corresponds to a lower learning rate and higher estimation error (See Appendix F.5 for further discussion with a toy example.)

**Asymptotic Normality.** Our kernel selection criterion for KCI is based on the asymptotic normality of U-statistics. We denote the KCI statistic $\|\Sigma_{\ddot{X}Y|Z}\|_{\mathrm{HS}}^2$ as $\mathrm{C}_{\mathrm{KCI}}^2$ for clarity. Then, we express $\mathrm{C}_{\mathrm{KCI}}^2$ as follows:

$$\mathrm{C}_{\mathrm{KCI}}^2 = \mathbb{E}\left[k_{\mathcal{Z}}(z,z') r_{x|z}(s,s') r_{y|z}(s,s')\right], \qquad (4)$$

where $s$ and $s'$ represents two different and independent copies of $(X,Y,Z)$ with $s = (x,y,z)$. $r_{x|z}(s,s')$ is the inner product of regression residuals given by $r_{x|z}(s,s') = \left\langle \phi_x(x) - \mu_{X|Z}(z), \phi_x(x') - \mu_{X|Z}(z') \right\rangle$ and similarly for $r_{y|z}(s,s')$.

Suppose we have $n$ i.i.d. observational points $S = \{s_i\}_{i=1}^n$ with $s_i = (x_i, y_i, z_i)$ being the one sample of $(X,Y,Z)$. We can intuitively give an unbiased U-statistic estimator for $\mathrm{C}_{\mathrm{KCI}}^2$, given by:

$$\hat{\mathrm{C}}_{\mathrm{KCIu}}^2 = (n)_2^{-1} \sum_{i,j \neq i} \hat{h}_{ij}, \quad \text{and} \quad \hat{h}_{ij} = k_{\mathcal{Z}}(z_i, z_j) \hat{r}_{x|z}(s_i, s_j) \hat{r}_{y|z}(s_i, s_j), \qquad (5)$$

where $\hat{r}_{x|z}(s_i, s_j) = \langle \phi_x(x_i) - \hat{\mu}_{X|Z}(z_i), \phi_x(x_j) - \hat{\mu}_{X|Z}(z_j) \rangle$ and $\hat{r}_{y|z}(s_i, s_j)$ are the estimated residuals with estimated $\hat{\mu}_{X|Z}$ and $\hat{\mu}_{Y|Z}$ [2]. $\hat{C}^2_{\text{KCIu}}$ has expectation zero under the null hypothesis $H_0$: $X \perp\!\!\!\perp Y \mid Z$, and has a positive expected value under $H_1$.

For a sufficiently large sample size $n$, $\hat{C}^2_{\text{KCIu}}$ can be considered as the asymptotic average of independent and identically distributed random variables. Based on the properties of U-statistics (see, e.g., Lee [2019, Section 3.2.1]), the asymptotic distribution of $\hat{C}^2_{\text{KCIu}}$ can be given by the Central Limit Theorem. If $\mathbb{E}(h^2) < \infty$ (which holds true for bounded continuous kernels): under the alternative hypothesis $H_1$, where $X \not\perp\!\!\!\perp Y \mid Z$, we have:

$$\sqrt{n}\left(\hat{C}^2_{\text{KCIu}} - C^2_{\text{KCI}}\right) \xrightarrow{d} \mathcal{N}(0, 4\sigma_1^2), \tag{6}$$

where $\sigma_1^2 = \text{Var}[h_1(s)]$ is the asymptotic variance with $h_1(s) = \mathbb{E}_{s'}[k_{\mathcal{Z}}(z, z')r_{x|z}(s, s')r_{y|z}(s, s')]$. With the fact that $\mathbb{E}_s[h_1(s)] = C^2_{\text{KCI}}$, we can derive that

$$\sigma_1^2 = \text{Var}[h_1(s_i)] = \mathbb{E}_{s_i}[(h_1(s_i) - \mathbb{E}_{s_i}[h_1(s_i)])^2] = \mathbb{E}_{s_i}[\mathbb{E}_{s_j}[h_{ij}] - C^2_{\text{KCI}}]^2. \tag{7}$$

**Test Power.** Based on the asymptotic normality in Eq. 6, we can estimate the test power, which represents the probability of correctly rejecting $H_0$ when $H_1$ is true for a given case. For large enough $n$, assuming that the conditional expectations are well estimated, the power is thus, using $\text{Pr}_1$ to denote the probability under $H_1$,

$$
\begin{aligned}
\text{Pr}_1\left(n\hat{C}^2_{\text{KCIu}} > r\right) &= \text{Pr}_1\left(\frac{n(\hat{C}^2_{\text{KCIu}} - C^2_{\text{KCI}})}{2\sqrt{n}\sigma_1} > \frac{r - nC^2_{\text{KCI}}}{2\sqrt{n}\sigma_1}\right) \\
&\to \Phi\left(\frac{\sqrt{n}C^2_{\text{KCI}}}{2\sigma_1} - \frac{r}{2\sqrt{n}\cdot\sigma_1}\right),
\end{aligned}
\tag{8}
$$

where $\Phi$ is the CDF of the standard normal distribution and $r$ is the rejection threshold, which is a constant for a specified significance level. The test power therefore can be maximized by maximizing the argument in $\Phi$. Since $C^2_{\text{KCI}}$ and the asymptotic variance $\sigma_1$ are also constant, for reasonable large sample size $n$, the power is asymptotically dominated by the first term, i.e. $\sqrt{n}C^2_{\text{KCI}}/2\sigma_1$.

**Kernel selection.** Following [Gretton et al., 2012b, Liu et al., 2020, Sutherland et al., 2021], we adopt the ratio of $C^2_{\text{KCI}}$ to $\sigma_1$ as our criterion, which asymptotically estimates the test power for the given kernel parameters. Both $C^2_{\text{KCI}}$ and $\sigma_1$ depend not only on the underlying distribution but also the kernel parameters. In practice, we use their empirical estimators from training samples to estimate the test power under the given kernels:

$$\hat{J}(S, \omega) = \hat{C}^2_{\text{KCIu},\omega}/\hat{\sigma}_{1,\omega}, \tag{9}$$

where $\omega$ denotes the kernel parameters to be selected, and $\hat{\sigma}_{1,\omega}$ is the estimated asymptotic variance:

$$\hat{\sigma}_{1,\omega}^2 = \frac{1}{n}\sum_i[h_1(s_i) - \hat{C}^2_{\text{KCIu},\omega}]^2 = \frac{1}{n}\sum_i\left[\left(\frac{1}{n-1}\sum_{j\neq i}\hat{h}_{ij}\right) - \hat{C}^2_{\text{KCIu},\omega}\right]^2. \tag{10}$$

We evaluate this criterion across a candidate set of kernel parameters and select the one that maximizes $\hat{J}(S, \omega)$, which corresponds to the highest estimated test power.

**Grid search-based kernel selection.** We adopt grid search for kernel parameter selection rather than continuous optimization. Specifically, we predefine a candidate set of kernel parameters, estimate the test power for each using Eq. 9, and select the one that maximizes the criterion for the final CI test. This design is motivated by the unique challenges of the CI testing setting. On one hand, KCIT involves estimating CMEs, which introduce intrinsic model estimation bias. This bias renders gradient-based optimization unreliable for improving test power, as convergence is not guaranteed—a limitation we also verify empirically in Appendix F.4. Consequently, power-based continuous optimization fails to identify optimal kernel parameters, undermining one of its main advantages.

---

[2]One may also consider using an HSIC-like unbiased estimator [Song et al., 2012, Theorem 5]; however, it is more complex and analytically intractable. This added complexity arises from the centralization of the kernel matrix in HSIC, which is unnecessary for KCI since the residuals are already estimated to have zero mean.

Nevertheless, while gradient information may be unreliable, the criterion itself remains informative. In particular, we show theoretically that under a sufficiently large sample size, the proposed criterion can still reliably rank candidate kernel parameters, even when continuous optimization is infeasible. On the other hand, continuous optimization in CI testing involves repeated regression estimation, resulting in significant computational cost and rendering it impractical for real-world applications. In contrast, grid search naturally supports parallelization, providing a more tractable alternative. We therefore adopt grid search as a practical and effective strategy for kernel selection in CI testing. The complete procedure is detailed in Appendix B.

## 4 Theoretical Result

In this section, we establish the convergence properties of the proposed kernel selection criterion. Since CI testing indirectly measures the dependence between residuals, it is essential to consider both the regression-induced estimation bias introduced by regressions and the random error resulting from finite data.

Thus, we impose the following assumptions on the boundedness of the kernels and CME estimators:

- **(Boundedness)** Under the kernel parameters $\omega = (\omega_x, \omega_y, \omega_z)$, the kernel $k_{\omega_z}$, the residuals $r_{X|Z}^{\omega_x}$, $r_{Y|Z}^{\omega_y}$ and their empirical counterparts $\hat{r}_{X|Z}^{\omega_x}, \hat{r}_{Y|Z}^{\omega_y}$ are bounded:

$$\sup_{(x,z)\in(\mathcal{X},\mathcal{Z})} \left| r_{X|Z}^{\omega_x} \right| \leq \nu, \qquad \sup_{(y,z)\in(\mathcal{Y},\mathcal{Z})} \left| r_{X|Z}^{\omega_y} \right| \leq \nu, \qquad \sup_{z\in\mathcal{Z}} k_{\omega_z} \leq \nu_z,$$

$$\sup_{(x,z)\in(\mathcal{X},\mathcal{Z})} \left| \hat{r}_{X|Z}^{\omega_x} \right| \leq \nu, \qquad \sup_{(y,z)\in(\mathcal{Y},\mathcal{Z})} \left| \hat{r}_{X|Z}^{\omega_y} \right| \leq \nu.$$

  Then $h_{ij}$ and $\hat{h}_{ij}$ are bounded within $[-\nu_z \nu^2, \nu_z \nu^2]$.

- **(EVD)** $(\mu_i)_{i\in I}$ are the eigenvalues of the operator $C_{ZZ} = \mathbb{E}[\phi_z(Z) \otimes \phi_z(Z)]$ in the kernel ridge regression. For some $c_\mu > 0$ and $p \in (0, 1]$ and for all $i \in I$,

$$\mu_i \leq c_\mu i^{-1/p}.$$

- **(SRC)** There exists $1 < \beta \leq 2$ such that

$$\mu_{X|Z} \in [\mathrm{HS}_{xz}]^\beta, \quad \mu_{Y|Z} \in [\mathrm{HS}_{yz}]^\beta,$$

  where $[\mathrm{HS}]^\beta$ denotes the interpolation space of the original HS space (also written as $[\mathrm{HS}]^1$), the eigenvalues decay of the functions in $[\mathrm{HS}]^\beta$ is lower bounded by $\beta$.

Assumption (Boundedness) imposes boundedness constraints on the kernel function and residual terms, which are mild and typically satisfied under appropriately chosen kernels and standard regression algorithms. Assumptions EVD and SRC are standard in vector-valued regression theory [Fischer and Steinwart, 2020, Li et al., 2022], and are widely used to characterize the convergence behavior of conditional mean regressors. Following [Pogodin et al., 2024], we focus on the well-specified cases with $\beta \in (1, 2]$, and the corresponding Hilbert space norm is well-defined. The parameter $\beta$ reflects the smoothness of the regression functions, with larger values indicating higher regularity. The choice of $\omega_x$ and $\omega_y$ in $\omega$ will affect the corresponding value of $\beta$. The following theorem provides the convergence rate of the proposed criterion.

**Theorem 3.** *Assuming the above assumptions hold. Let $\bar{\Omega}_c$ be the subset of kernel parameters of $\omega$ such that the asymptotic variance $\sigma_{1,\omega}^2 \geq c^2$ for some constant $c > 0$. Then, for any $\omega \in \bar{\Omega}_c$, with probability at least $1 - \delta$, we have a constant $K > 0$ independent of $n$ and $\delta$ that:*

$$\left| \frac{\hat{\mathrm{C}}_{\mathrm{KCIu},\omega}^2}{\hat{\sigma}_{1,\omega}} - \frac{\mathrm{C}_{\mathrm{KCI},\omega}^2}{\sigma_{1,\omega}} \right| \sim \mathcal{O}\left( C_1 n^{-\frac{\beta_\omega - 1}{2(\beta_\omega + p)}} + C_2 n^{-\frac{1}{2}} \right) \tag{11}$$

*where $C_1 = (\frac{4}{c} + \frac{64\nu_z^2\nu^4}{3c^3})\nu_z\nu\sqrt{\nu K}\ln\frac{4}{\delta}$ and $C_2 = (\frac{2}{c} + \frac{8}{3c^3})\nu_v\nu^2\sqrt{2\ln\frac{2}{\delta}}$.*

The complete proof is provided in Appendix D, where we decompose the total error of the statistic estimator into two components: the regression-induced estimation bias and the random error arising from finite sampling, assuming accurate regression. The result shows that, for fixed kernel parameters, the empirical criterion converges to its population counterpart, with a convergence rate governed by both components. The first term reflects the regression bias, which is influenced by both the smoothness of the true conditional mean functions and the choice of kernel parameters. The second term corresponds to the random variation due to finite-sample effects. The appearance of the bias term is intrinsic to CI testing, which evaluates dependence between residuals—unlike standard testing problems [Gretton et al., 2012b, Albert et al., 2022], where only variance terms are typically considered. Nevertheless, for a reasonably large sample size, the empirical criterion remains sufficiently close to the population counterpart, providing a reliable basis for comparing test power across different kernel choices.

## 5 Experimental Results

In this section, we empirically analyze our Power method and its variant on CI tests using synthetic and real benchmarks, compare it with baseline methods, and assess its impact on causal discovery.

**Implementation details.** We use Gaussian kernels for all the kernels involved. For the kernel parameters of $\phi_x$ and $\phi_y$, we use the median heuristic as the initial value and apply different weights. Specifically, we take the median heuristic as a sensible initialization and use the candidate weight list $[0.1, 0.3, 0.75, 0.88, 1, 1.25, 1.5, 3, 5, 10]$, applying each weight as a multiplier to the median-based bandwidth. For $K_z$ in the statistic, we decide it using the median heuristic without further selection. That is, our method considers $10 \times 10 = 100$ possible parameter combinations. We evenly divide all samples into a training set and a testing set. During the testing phase, we use the weighted sum of chi-squared to compute $p$-value. The significance level is set to the default value of 0.05. Please refer to Appendix E.1 for more implementation details.

### 5.1 Synthetic Data

In the synthetic experiment, we assume $X$ and $Y$ are dependent variables conditioned on $Z$. We analyzed our method's performance under varying dimensions of the conditioning set $Z$ and different sample sizes. To examine Type I errors, we generated $X$ and $Y$, which should be independent given $Z$, using the following nonlinear additive functional model:

$$X = f(W^\top Z) + \gamma E, \tag{12}$$

where $W \sim \mathcal{N}(0, I_{d_z})$ and $d_z$ represents the dimension of $Z$, $f$ was randomly chosen from the *linear*, *sin*, *cos*, $x^2$, $2^x$ and $\exp(x)$. When $f$ is neither *sin* nor *cos*, an additional $1/\sqrt{d_z}$ is multiplied to $f(W^\top Z)$ to balance the scale of the function and noise. The noise $E$ was randomly chosen from either a *Gaussian* or *uniform* distribution with the noise scale $\gamma = 1$. To examine Type II errors, we added an additional variable $T$ to both $X$ and $Y$, making them conditionally dependent given $Z$. $T \sim \mathcal{N}(0, 1)$, and

$$X = f(W^\top Z) + E + \alpha T, \tag{13}$$

with $\alpha = 0.5$. For each setting, we randomly repeated the process 1000 times to obtain Type I and Type II error. For further implementation details, please refer to Appendix E.2.

#### 5.1.1 Comparison with baseline methods

**Baselines.** We first compare our proposed power-based method with CI baselines. Our proposed method is denoted as *Power*, while the median heuristic-based method is denoted as *Median*. In Median, we still decompose $\phi(x, z)$ into $\phi(x) \otimes \phi(z)$, and the kernel bandwidth is determined by the median heuristic without selection. In Power, we perform a grid search to select the kernel bandwidths involved in $\phi_x$ and $\phi_y$, choosing the parameters with the highest criterion for testing on the test data. We further compare it with the kernel-based CIRCE [Pogodin et al., 2022], which only considers the independence between one-sided residuals and the other dependent variable itself. Additionally, we compare with regression-based methods GCM[Shah and Peters, 2020] and RBPT2 [Polo et al., 2023], which conduct the regression in $L^2$ space. (See Appendix E.4 for more details about CIRCE, GCM and RBPT2).

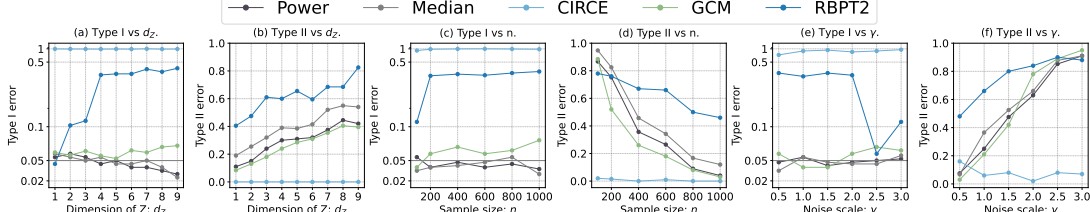

Figure 1: Performance comparison on synthetic data. Left: Type I error (a) and Type II error (b) when increasing the dimension of $Z$ $d_Z$, keeping the sample size $n = 500$ and noise scale $\gamma = 1$. Middle: Type I error (c) and Type II error (d) when increasing samples size, keeping $d_Z = 5$ and $\gamma = 0.5$. Right: Type I error (e) and Type II error (f) across different noise scales $\gamma$ ($d_Z = 5$, $n = 500$).

**On the dimension of $Z$.** Figure 1(a) and (b) illustrate the performance of Power and baseline methods, with an increasing the dimension of $Z$. In terms of Type I error control, RBPT2 and CIRCE failed to maintain Type I error below the given significance level, while GCM slightly exceeds the given significance level. KCI-based Median and Power successfully controlled it, demonstrating the robustness of KCI. Regarding Type II error, RBPT2 exhibited a slight increase in Type II error as the dimension of $Z$ increased, performing almost like random guessing in general. CIRCE accepted all instances of $H_1$, which consistent with the performance on Type I error. Both Median and Power exhibit an increasing Type II error as $d_z$ increases. However, Power maintains a consistently lower Type II error than Median by a substantial margin. With a similar lower Type I error, it demonstrate that Power can effectively select more proper kernel parameters with higher test power while maintaining Type I error control across different dimensions setting. The above results reflect the robustness of KCI in controlling the Type I error across a range of dependency structures, as well as the improvement of our Power over the median heuristic Median.

**On the sample size.** We also evaluated the performance by varying the sample size $n$, shown in Figure 1(c) and (d). The performance of GCM and CIRCE remains consistent with previous analysis, showing no clear convergence trend as the sample size increases. Meanwhile, the Type II error of RBPT2 decreases as $n$ increases, but its Type I error remains uncontrolled. In contrast, our Power consistently outperforms the median heuristic-based Median across different sample sizes, and the gap between them does not diminish as $n$ increases.

**On the noise scale.** In Figure 1(e) and (f), we also analyzed the performance under different noise scales $\gamma$, with the scale of the latent variable $T$ fixed at $\alpha = 0.5$ (in Eq. 49). From the results, we observe that when the noise scale is relatively low, Power struggles to achieve further improvements through kernel selection. Conversely, when $\gamma$ is too large, making it difficult to separate $T$ from the noise variable, Power also loses its effectiveness. This indicates that the improvement of Power over Median primarily stems from its ability to handle the overlapping region between $H_0$ and $H_1$, where different parameter choices correspond to different levels of overlap. However, when the overlap is either too small ($\gamma$ is very low) or too large ($\gamma$ is too high), the region that can be adjusted by kernel parameters becomes insignificant, reducing the effectiveness of our method.

Due to space constraints, additional results are provided in the appendix: experiments on a real-world conditional independence benchmark (car insurance dataset) in Appendix F.1.1, extended baseline comparisons and results on non-additive synthetic data in Appendix F.2, and evaluations under high-dimensional conditioning settings in Appendix F.3.

### 5.1.2 Ablation Study

In this section, we take a closer look at our method by studying its variants, with the results presented in Figure 2.

**On the selectable kernels.** In Figure 2(a) and (b), we examine the impact of kernel selection on different components by selecting only $\phi(z)$ in the statistic (but not in the regression), denoted as *SelZ*, while fixing $\phi(x)$ and $\phi(y)$ using the median heuristic. Similarly, we consider selecting only $\phi(y)$, denoted as *SelY*. *SelZ* does not improve the Type II error compared to *Median*, whereas *SelY* reduces the Type II error relative to *Median* but still underperforms *Power*. This suggests that one-sided selection is somewhat effective, but joint selection, as in *Power*, provides the best performance.

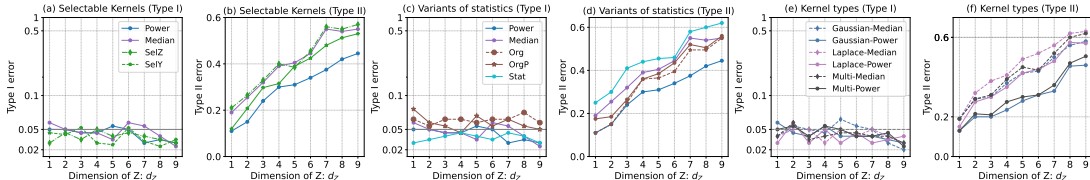

Figure 2: Ablation study on the dimension of $Z$ with $n = 500$ and $\gamma = 1$. (a) and (b) - Different choices on the selectable kernels. (c) and (d) - Different variants of the statistic and selection criterion. (e) and (f) - Different kernel types.

**On the decomposition.** Additionally, in Figure 2(c) and (d), we compare against the original form of KCI without decomposing $\phi(x, z)$ (i.e., Eq. 2), denoted as *Org*. The kernel parameters in *Org* are determined using the median heuristic without selection. We also consider a variant, denoted as *OrgP*, which applies the proposed kernel selection strategy to both $\phi_{xz}$ and $\phi_y$ in *Org*. In terms of Type I error, *Org* and *OrgP* exceed the significance level, indicating poor error control. While *Org* achieves slightly lower Type II error than *Median*, this suggests it tends to over-detect conditional dependence due to insufficient removal of $Z$'s influence. The weaker CME performance likely results from slower convergence, consistent with Appendix F.5. This highlights that using the original KCI without decomposition introduces notable estimation bias, affecting both test outcomes and the accuracy of the proposed criterion.

**On the criterion.** Fukumizu et al. [2009] proposed directly maximizing the statistic for kernel selection. We follow their approach and perform kernel selection directly based on the statistic itself, denoted as *Stat*. The results in Figure 2(c) and (d) show that Stat performs even worse than Median, with a larger Type II error, highlighting the importance of selecting kernel parameters that need to simultaneously consider minimizing the asymptotic variance.

**On the kernel types.** In Figure 2 (e) and (f), we compare the performance of using different kernel types individually as well as jointly. Our method can also be applied with other suitable kernels or used to select among different kernel classes. According to the definition of the cross-covariance operator, $k_X$ and $k_Y$ in KCIT need to be characteristic kernels [Fukumizu et al., 2007]. Therefore, we conduct experiments using Gaussian and Laplace kernels. We use the same median heuristic to initialize the bandwidth in the Laplace kernel, along with the same power-based selection strategy. Our Power are also able to perform selection over multiple kernel types (with median-heuristic initialization and fixed, denoted as *Multi-Median*) as well as over the parameters selectable version, referred to as *Multi-Power*. We observe that for both Laplace and Gaussian kernels, the power-based kernel selection effectively reduces the Type II error. The performance of Multi-Power is slightly worse than that of Gaussian-Power, suggesting that the Gaussian kernel is generally a strong choice and performs better than the Laplace kernel in most cases. The observed performance drop may be attributed to estimation errors in the power computation. Overall, our method demonstrates the ability to effectively select parameters across multiple kernel types.

Table 1: Average testing time (s) ± standard deviation on different sample sizes.

| Sample Size | 200 | 400 | 600 | 800 | 1000 |
|---|---|---|---|---|---|
| Power | 0.595±0.02 | 1.758±0.14 | 3.869±0.60 | 9.641±2.37 | 10.12±6.54 |
| Median | 0.302±0.06 | 0.914±0.14 | 1.742±0.32 | 4.422±1.99 | 4.358±2.45 |

### 5.1.3 Testing Time

Power involves more regression learning with different parameters; however, due to the grid-based search and the independence of different regressions, it enables efficient parallel training. Table 1 presents the overall runtime of Power compared to Median for different sample sizes. It shows that the testing time of Power is approximately twice that of Median. Notably, this additional computational cost does not increase linearly with the number of searches but considers over 100 parameter combinations, making it manageable and ensuring that our method remains practical for real-world applications. For more details, please refer to Appendix E.3.

## 5.2 Comparison on causal discovery

Our method can also be directly extended to causal discovery tasks [Glymour et al., 2019, Zhang et al., 2018]. Formally, given a set of observations of random variables, causal discovery methods seek to depict the causal relationships among these variables through a directed acyclic graph (DAG). CI testing is central to constraint-based pipelines [Pearl and Mackenzie, 2018]: early CI outcomes drive adjacency pruning and orientation rules, so inaccuracies can propagate and reshape the inferred graph. The overall fidelity of these methods therefore hinges on CI procedures that are reliable and stable across varying conditioning sets and sample sizes.

We compared the performance of Power with Median using the PC algorithm[Spirtes et al., 2000] as the search method. We generated the synthetic causal graphs with varying graph densities ranging from 0.2 to 0.8. Each generated graph involves 10 variables with sample sizes of $n = 500$. For each variable $X_i$ in the graph, the data was generated according to $X_i = f_i(W_i^\top \mathrm{PA}_i) + E_i$, where $\mathrm{PA}_i$ are parent nodes of $X_i$ in the graph, $W_i \sim \mathcal{N}(0, I_{d_{PA}})$ and $E_i$ is the noise term and $f_i$ was randomly selected. Additionally, we conducted experiments on the real-world causal discovery benchmarks SACHs [Sachs et al., 2005] and CHILD [Spiegelhalter et al., 1993]. We evaluate our Power and Median using F1 score and a higher F1 score indicates greater accuracy From Table 2, Power outperforms Median in most graph density settings, particularly on denser graphs. Power also outperforms the median heuristic on both SACHs and CHILD. This indicates that our proposed kernel selection method, Power, can benefit causal discovery tasks in general. For more implementation details and results on SACHs and CHILD, please refer to Appendix E.1 and Appendix F.1.2.

Table 2: F1 score on synthetic graph and on the real world benchmarks. Bold represents the better.

| Graph Density | 0.2 | 0.3 | 0.4 | 0.5 | 0.6 | 0.7 | 0.8 | SACHs | CHILD |
|---|---|---|---|---|---|---|---|---|---|
| Power | 0.656 ±0.057 | **0.637** ±0.062 | **0.603** ±0.047 | **0.581** ±0.052 | **0.567** ±0.044 | **0.515** ±0.032 | **0.461** ±0.045 | **0.674** ±0.032 | **0.762** ±0.052 |
| Median | **0.657** ±0.067 | 0.623 ±0.077 | 0.586 ±0.032 | 0.548 ±0.045 | 0.523 ±0.042 | 0.490 ±0.043 | 0.443 ±0.037 | 0.576 ±0.022 | 0.790 ±0.044 |

## 6 Conclusion and Future Work

In this paper, we propose a practical kernel selection method for KCI, replacing the coarse median heuristic. To address the model estimation bias inherent in CI testing, we decompose the KCI statistic and perform parameter selection via grid search based on the estimated test power. We provide a convergence analysis of the proposed criterion that explicitly accounts for estimation error, and experimental results validate the effectiveness of our approach.

Currently, the search procedure relies on a fixed grid, which does not guarantee optimal parameter selection. Moreover, the current theoretical framework does not yet provide formal Type I error control for KCI under general conditions, unlike results established in more restrictive settings such as additive models [Shah and Peters, 2020, Niu et al., 2024]. This limitation stems from the fact that KCI addresses general CI relationships, leading to a more tractable null distribution with kernel components. And the null distribution of KCI remains complex and sensitive to estimation bias. A promising future direction is to establish uniform convergence results over kernel parameters and derive the conditions and convergence rates for valid Type I error control. Such developments would pave the way for a more flexible and theoretically grounded adaptive kernel selection strategy: bringing KCI closer to achieving optimal kernel choice for CI testing.

## Acknowledgements

We would like to acknowledge support from ARC (DP240102088, DP230101540, DE240101089, LP240100101), the NSF (DMS-2428058; Award No. 2229881), the RGC (Young Collaborative Research Grant No. C2005-24Y; General Research Fund No. 12200725), the AI Institute for Societal Decision Making (AI-SDM), the National Institutes of Health (R01HL159805), and grants from Quris AI, Florin Court Capital, the MBZUAI-WIS Joint Program, and the Al Deira Causal Education project, as well as the NSF&CSIRO Responsible AI program (Grant No. 2303037).

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

# Appendices

In the appendix, we provide additional discussions on related work, the assumptions required for the theoretical guarantees, complete proofs of the main results, as well as further experiments and analysis. The structure is as follows:

- Appendix A provides an overview of related work relevant to our proposed method, including recent developments in kernel-based conditional independence testing.
- Appendix B describes the training and testing procedures used:
    - Appendix B.1: The kernel selection and test procedure overview.
    - Appendix B.2: Details of the testing pipeline.
- Appendix C presents the theoretical assumptions required to ensure the validity of our convergence results and the consistency of the proposed criterion.
- Appendix D contains the complete proof of the convergence rate (Theorem 3), including:
    - Appendix D.1: The convergence rate of $\hat{\eta}$.
    - Appendix D.2: The convergence of $\hat{\sigma}$.
    - Appendix D.3: Supporting results on conditional mean embedding (CME) convergence.
- Appendix E provides implementation and experimental setup details, including:
    - Appendix E.1: Model hyperparameter settings.
    - Appendix E.2: Synthetic data generation procedures.
    - Appendix E.3: Runtime benchmarking configurations.
    - Appendix E.4: Description of conditional independence testing baselines.
- Appendix F includes additional experimental results and discussions beyond the main paper:
    - Appendix F.1: Real-data evaluation,
        * Appendix F.1.1: Results on the car insurance dataset.
        * Appendix F.1.2: Causal discovery benchmarks.
    - Appendix F.2: More results on synthetic datasets.
    - Appendix F.3: Analysis under high-dimensional conditioning.
    - Appendix F.4: Comparison with continuous kernel parameter optimization.
    - Appendix F.5: Toy example illustrations.

# A   Related work

**Conditional independence testing.** Conditional independence (CI) testing serves as a fundamental building block in many machine learning and statistical inference tasks, especially causal discovery [Zhang et al., 2011, Cai et al., 2022, Liu et al., 2024, Lin et al., 2024]. There is a growing body of literature on conditional independence test, which can be roughly divided into three groups: (1) regression-based methods [Shah and Peters, 2020, Polo et al., 2023, Scheidegger et al., 2022, He et al., 2021]; (2) simulation-based methods [Doran et al., 2014, Candes et al., 2018, Berrett et al., 2020, Li et al., 2023, Sen et al., 2017, Zhang et al., 2024] and (3) kernel-based methods [Zhang et al., 2011, Fukumizu et al., 2007, Kour and Saabne, 2014, Scetbon et al., 2022].

Regression-based methods require assumptions about the relationship and noise structure, as well as the assumptions of removal of any information from the conditioning set $Z$ by regression. When these assumptions hold, regression-based methods have been shown to effectively control Type I error; otherwise, they do not. Another important category is simulation-based methods (also known as randomization-based methods), which primarily implicitly or explicitly approximate the conditional distributions $P_{X|Z}$ or $P_{Y|Z}$ to simulate the null distribution. A clear drawback is that such approaches often come with significant approximation errors, leading to an inflation of the type-I error and rendering the test invalid.

Kernel-based CI methods, on the other hand, do not require additional assumptions and can detect general dependence. By mapping variables into a RKHS, kernel functions enable the assessment of similarities between high-dimensional implicit functions, thereby capturing higher-order statistical moments. Utilizing characteristic kernels allows us to infer distribution properties such as homogeneity [Gretton et al., 2012a], independence [Gretton et al., 2005b], and conditional independence [Fukumizu et al., 2007, Sun et al., 2007, Zhang et al., 2011, Huang et al., 2022]. These properties make kernel-based methods capable of discerning conditional independence in CI tasks without the need to simulate intricate conditional distributions.

**The hardness of CI testing.** Due to the unique nature of CI testing, Shah and Peters [2020] demonstrated that a valid CI test does not have power against any alternatives. This implies that no method can simultaneously control the Type I error rate at the given significance level while maintaining adequate power. This is due to the inherent nature of the CI testing task, which requires considering conditional distributions that are difficult to directly obtain and estimate from observed samples. As a result, CI testing inherently requires accounting for estimation errors, whereas other hypothesis testing tasks do not. Consequently, reducing estimation error is crucial for CI testing.

Based on this challenge, several assumptions have been proposed. The Model-X assumption [Candes et al., 2018] assumes that one side of the conditional distribution (or its likelihood) is known exactly. Another class of approaches, known as doubly robust methods [Shah and Peters, 2020], imposes stricter requirements on the convergence rate of estimation errors to ensure reliable testing. The kernel-based KCI can currently only achieve pointwise Type I error control, and its underlying assumptions require further investigation. In this paper, we decompose the original KCI formulation, which has been shown to reduce the estimation error of residuals, thereby improving the validity and reliability of the test.

**Kernel selection.** Although kernel-based methods are capable of detecting general dependencies between variables, a critical aspect that significantly influences their effectiveness is the choice of kernel functions. This selection process primarily focuses on tuning kernel parameters, such as the bandwidth in radial basis function (RBF) kernel, which can often be more influential than the choice of the kernel family [Schölkopf et al., 2002, Section 4.4.5]. Most existing kernel-involved methods rely on heuristic parameter selection, typically choosing the bandwidth of Gaussian kernels based on the median of pairwise distances in the data, a strategy commonly referred to as the median heuristic [Schölkopf et al., 2002]. [Kim et al., 2006] The selection of appropriate kernels remains an unresolved question in numerous studies [Chu and Marron, 1991, Herrmann et al., 1992, Chapelle and Vapnik, 1999, Kim et al., 2006]. Fukumizu et al. [2009] propose selecting kernel parameters by directly maximizing the MMD statistic itself, which is shown to be equivalent to minimizing the classification error under a linear loss. However, Gretton et al. [2012b] argue that this approach is suboptimal, as it neglects the variance component of the test statistic. They instead propose using a statistic-to-variance ratio criterion as a surrogate for the estimated test power under the current kernel parameters.

Variants of this criterion have been further developed and applied to tasks such as two-sample testing [Liu et al., 2021, Biggs et al., 2023], unconditional independence testing [Albert et al., 2022, Ren et al., 2024a,b], and goodness-of-fit testing [Schrab et al., 2022]. These methods support continuous optimization of kernel parameters and are capable of selecting near-optimal or even oracle kernel parameters. Furthermore, deep kernels parameterized by neural networks have also been proposed and integrated into these testing tasks, as in [Liu et al., 2020, Xu et al., 2024]. While these advances have shown promise, they are not directly applicable to CI testing task. Both two-sample testing and unconditional independence testing directly measure distributional differences between variables, where the main source of error typically arises from finite-sample variability. In contrast, CI testing is substantially more difficult, as it requires accounting for the conditional distributions, which may act as latent sources of dependence. This often necessitates residualization procedures to remove the conditional effect, as implemented in methods such as KCI or GCM [Shah and Peters, 2020]. Therefore, for CI task, it possesses distinct characteristics, primarily involving the consideration of regression residuals, which inherently contain biases. And the choice of kernel is intuitively crucial for kernel-based CI methods, but it has remained an open issue.

Both two-sample testing and unconditional independence testing directly assess distributional differences between variables, where the primary source of error typically arises from finite-sample variability. In contrast, CI testing is substantially more challenging, as it involves reasoning about

conditional distributions, which can act as latent sources of dependence. To address this, residualization procedures are often employed to remove the conditional effect, after which conditional independence is assessed based on the resulting residuals, as in methods such as KCI or GCM [Shah and Peters, 2020]. As a result, CI testing presents distinct challenges—most notably, the reliance on regression residuals, which inherently introduce estimation bias. While the choice of kernel is intuitively critical for the performance of kernel-based CI methods, identifying principled strategies for kernel selection remains an open and unresolved problem.

# B   Procedure Details

## B.1   Overall procedure

---

**Algorithm 1:** Overall Procedure of *Power*

---

**Input:** Observations of $(X, Y, Z)$ for training $S_{tr} = \{(x_i, y_i, z_i)\}$ and for testing $S_{te}$; kernel
types for $\phi_x, \phi_y, \phi_z$; candidate parameter weights.
**Output:** Test decision of $X \perp\!\!\!\perp Y \mid Z$.
// Step 1:   Initialize kernel parameters
1  Use median heuristic to compute initial kernel widths $\omega_x, \omega_y, \omega_z$;
2  Generate candidate lists for $\omega_x$ and $\omega_y$ by applying predefined weight multipliers;
3  Fix $\omega_z$ as the median heuristic value (no selection);
// Step 2-4 can be computed independently and thus parallelized
4  **foreach** *parameter combination* $\omega_i = (\omega_x, \omega_y)$ *in parallel* **do**
    // Step 2:   Compute conditional embeddings
5      Compute kernel matrices $K_X, K_Y, K_Z$;
6      Estimate $\mu_{X|Z} = K_Z^R (K_Z^R + \varepsilon I)^{-1} \phi_x(X)$;
7      Estimate $\mu_{Y|Z} = K_Z^R (K_Z^R + \varepsilon I)^{-1} \phi_y(Y)$;
    // Step 3:   Compute residual matrices
8      Compute projection operator $R_Z = \varepsilon (K_Z^R + \varepsilon I)^{-1}$;
9      Compute $R_{X|Z} = R_Z K_X R_Z$, $R_{Y|Z} = R_Z K_Y R_Z$;
    // Step 4:   Evaluate criterion
10     Compute $\hat{J}(S, \omega_i)$ as in Eq. 9;

// Step 5:   Select optimal kernel parameters
11  Choose $\omega_m = \arg\max_{\omega_i} \hat{J}(S, \omega_i)$;
// Step 6:   Final testing
12  Perform the test using the selected parameters $\omega_m$ on test samples $S_{te}$;
13  Refer to Appendix B.2 for test procedure details;
14  **return** *Test result (reject/do not reject $H_0 : X \perp\!\!\!\perp Y \mid Z$)*;

---

## B.2   Testing procedure details

After selecting the kernel parameters with the maximum estimated test power on the training set, we follow the original KCI testing procedure to perform the testing process on the test set. With $m$ testing points, the KCI statistic $\mathrm{C}_{\mathrm{KCI}}^2$ has a biased HSIC-like estimator [Gretton et al., 2005b]:

$$\hat{\mathrm{C}}_{\mathrm{KCI}_b}^2 = \frac{1}{n(n-1)} \mathrm{Tr}[(K_Z^{te} \odot R_{X|Z}^{te}) R_{Y|Z}^{te}]. \tag{14}$$

We first compute the residual covariance matrices $R_{X|Z}^{te}$ and $R_{Y|Z}^{te}$ and the kernel matrix $K_Z^{te}$ with the selected $\omega_m$. Then, we denote $K = R_{X|Z}^{te} \odot K_Z^{te}$ and $L = K_{Y|Z}^{te}$ and let the EVD decomposition of $K$ and $L$ be $K = V_K \Lambda_K V_K$ and $L = V_L \Lambda_L V_L$. $\Lambda_K$ (resp. $\Lambda_L$) is the diagonal matrix containing non-negative eigenvalues $\lambda_{K,i}$ (resp. $\lambda_{L,i}$). Let $\boldsymbol{\psi}_K = [\psi_{K,1}(\boldsymbol{x}), \cdots, \psi_{L,n}(\boldsymbol{x})] = V_K \Lambda_K^{1/2}$ and $\boldsymbol{\phi}_L = [\phi_{L,1}(\boldsymbol{y}, \boldsymbol{z}), \cdots, \phi_{L,n}(\boldsymbol{y}, \boldsymbol{z})] = V_L \Lambda_L^{1/2}$. And its null distribution can be approximated in two ways: as (1) weighted (infinite) sum of $\chi^2$ variables, or through (2) Gamma approximation.

**Weighted sum of $\chi^2$ approximation.** Under $H_0$, $X \perp\!\!\!\perp Y \mid Z$, $\hat{C}^2_{\mathrm{KCI}_b}$ has the same asymptotic distribution as

$$\check{T}_b = \frac{1}{n(n-1)} \sum_{k=1}^{n^2} \tilde{\lambda}_k \cdot z_k^2, \tag{15}$$

where $z_k \sim \mathcal{N}(0,1)$ and where $\tilde{\lambda}_k$ are eigenvalues of $\mathbf{w}\mathbf{w}^\top$ and $\mathbf{w} = [\mathbf{w}_1, \cdots, \mathbf{w}_n]$, with the vector $\mathbf{w}_t$ obtained by stacking $\boldsymbol{M}_t = [\psi_{K,1}(x_t), \cdots, \psi_{K,n}(x_t)]^\top \cdot [\phi_{L,1}(y_t, z_t), \cdots, \phi_{L,n}(y_t, z_t)]$. This conclusion primarily relies on the continuous mapping theorem, for details refer to [Zhang et al., 2011, Theorem 3].

**Gamma approximation.** Following [Gretton et al., 2007], the null distribution for the KCI estimator $\frac{1}{n^2}\mathrm{Tr}(KL)$ can also be approximated by a Gamma distribution, which is $p(t) = t^{k-1}\frac{e^{-t/\theta}}{\theta^k \gamma(k)}$, with the parameters

$$k = \frac{\mu}{\sigma^2}, \quad \theta = \frac{\sigma^2}{n\mu}, \quad \text{with} \quad \mu = \frac{1}{n}\mathrm{Tr}(\mathbf{w}\mathbf{w}^\top) \quad \text{and} \quad \sigma^2 = 2\frac{1}{n^2}\mathrm{Tr}[(\mathbf{w}\mathbf{w}^\top)^2]. \tag{16}$$

Therefore, one can use Monte Carlo simulation to approximate the null distribution according to the two approaches mentioned above. The complete testing procedure is as follows: we first estimate the conditional means $\mu_{X|Z}$ and $\mu_{Y|Z}$ and learn the parameters in $\phi_z$ on the training data and calculate the $K_{X|Z}$, $K_{Y|Z}$ and $K_Z$ on the testing data and the eigenvalues and eigenvectors of $K$ and $L$ defined above. Then we evaluate the statistic $\hat{C}^2_{\mathrm{KCI}_b}$ according to Equation 14. And then we simulate the null distribution either by (1) weighted sum of $\chi^2$ approximation (according to Eq. 15) or (2) Gamma approximation (with the parameters given by Eq. 16). We then obtain a set of statistics $\check{\boldsymbol{T}} = (\check{T}_1, \cdots, \check{T}_m)$ through sampling. Then the $p$-value is calculated as the proportion of the statistic $\check{T}_j$ in $\check{\boldsymbol{T}}$ that is greater than $\hat{C}^2_{\mathrm{KCI}_b}$. Finally, if the $p$-value is not greater than the given significance level $\alpha$, we reject $H_0$ and hold $H_1$; otherwise, we hold $H_0$.

## C  Assumptions

To analyze the convergence rate of the statistic, we require some assumption as follows under **certain fixed** kernel parameters $\omega = (\omega_x, \omega_y, \omega_z)$:

- The kernel $k_{\omega_z}$, the residuals $r_{X|Z}^{\omega_x}$, $r_{Y|Z}^{\omega_y}$ and their corresponding estimates $\hat{r}_{X|Z}^{\omega_x}$, $\hat{r}_{Y|Z}^{\omega_y}$ are uniformly bounded:

$$\sup_{(x,z)\in(\mathcal{X},\mathcal{Z})} |r_{X|Z}^{\omega_x}(x,x')| \le \nu, \quad \sup_{(y,z)\in(\mathcal{Y},\mathcal{Z})} |r_{X|Z}^{\omega_y}(y,y')| \le \nu, \quad \sup_{z\in\mathcal{Z}} k_{\omega_z}(z,z) \le \nu_z,$$
$$\sup_{(x,z)\in(\mathcal{X},\mathcal{Z})} |\hat{r}_{X|Z}^{\omega_x}(x,x')| \le \nu, \quad \sup_{(y,z)\in(\mathcal{Y},\mathcal{Z})} |\hat{r}_{X|Z}^{\omega_y}(y,y')| \le \nu. \tag{Boundedness}$$

   To simplify the notation, we denote $C_\nu = \nu_z \nu^2$. Consequently, $h_{ij}$ and $\hat{h}_{ij}$ are bounded in the range $[-C_\nu, C_\nu]$.

We require some additional assumptions regarding the CMEs involved for the analysis of the convergence of the CMEs involved ($\mu_{X|Z}$ and $\mu_{Y|Z}$). We directly adopt the framework and the regular assumptions from [Pogodin et al., 2024], which were originally introduced in [Fischer and Steinwart, 2020, Li et al., 2022]. We use the same names for the following assumptions to align with those in [Fischer and Steinwart, 2020].

- $(\mu_i)_{i\in I}$ is the eigenvalues of the operator $C_{ZZ} = \mathbb{E}[\phi_z(Z) \otimes \phi_z(Z)]$. For some $c_\mu > 0$ and $p \in (0,1]$ and for all $i \in I$,

$$\mu_i \le c_\mu i^{-1/p}. \tag{EVD}$$

- There exists $1 < \beta \le 2$ such that

$$\mu_{X|Z} \in [\mathrm{HS}_{xz}]^\beta, \quad \mu_{Y|Z} \in [\mathrm{HS}_{yz}]^\beta, \tag{SRC}$$

   where $[\mathrm{HS}]^\beta$ represents the interpolation space of the original space $\mathrm{HS}$ (also written as $[\mathrm{HS}]^1$), and the eigenvalues decay of the functions in $[\mathrm{HS}]^\beta$ is lower bounded by $\beta$.

In [Fischer and Steinwart, 2020, Li et al., 2022], there is another assumption *embedding property* (EMB), was introduced, which implies a polynomial eigenvalue decay of order $1/\alpha$ for $C_{ZZ}$. Since the EMB condition is always satisfied for $\alpha = 1$ and a bounded kernel $k_{\mathcal{Z}}$, it follows that, under Assumption Boundedness, the EMB condition is consistently satisfied within our framework. Therefore, we do not repeat further here.

# D  Proof of Convergence Rate

**Theorem 4.** *Assuming that $\omega_x$ and $\omega_y$ can parameterize uniformly bounded residuals $r_{X|Z}^{\omega_x}$ and $r_{Y|Z}^{\omega_y}$. Let $\bar{\Omega}_c$ be the set of kernel parameters of $\omega$ for which the asymptotic variance $\sigma_{1,\omega}^2 \geq c^2$ with a positive constant c. Under Assumptions (Boundedness), (EVD) and (SRC), and given that the kernel selection procedure exhibits density over $\bar{\Omega}_c$, then with probability at least $1 - \delta$,*

$$\left| \frac{\hat{C}_{\text{KCIu},\omega}^2}{\hat{\sigma}_{1,\omega}} - \frac{C_{\text{KCI},\omega}^2}{\sigma_{1,\omega}} \right| \sim \mathcal{O}\left( C_1 n^{-\frac{\beta-1}{2(\beta+p)}} + C_2 n^{-\frac{1}{2}} \right), \tag{17}$$

*where $C_1 = (\frac{4}{c} + \frac{64\nu_z^2\nu^4}{3c^3})\nu_z\nu\sqrt{\nu K}\ln\frac{4}{\delta}$ and $C_2 = (\frac{2}{c} + \frac{8}{3c^3})\nu_v\nu^2\sqrt{2\ln\frac{2}{\delta}} \cdot n^{-\frac{1}{2}}$.*

*Proof.* We denote $C_{\text{KCI},\omega}^2$ with the kernel parameters $\omega = (\omega_x, \omega_y, \omega_z)$ as $\eta_\omega$, its unbiased estimator $\hat{C}_{\text{KCIu},\omega}^2$ as $\hat{\eta}_\omega$ for clarity, Similarly, the asymptotic variance $\sigma_{1,\omega}$ is shorten as $\sigma_\omega$ with its corresponding regularized estimator $\hat{\sigma}_{1,\omega}$ as $\hat{\sigma}_\omega$. For reasonable large sample size $n_0$ which holds eq. (46), we have $\hat{\sigma}_\omega \geq \frac{c}{2}$ with high probability. Then with Assumption (Boundedness) that $|h_{ij}| \leq C_\nu$, we obtain $|\eta_\omega| = |\mathbb{E}[h_{ij}]| \leq C_\nu$ and $|\hat{\eta}_\omega| \leq C_\nu$. We begin by decomposing

$$\left| \frac{\hat{\eta}_\omega}{\hat{\sigma}_\omega} - \frac{\eta_\omega}{\sigma_\omega} \right| \leq \left| \frac{\hat{\eta}_\omega}{\hat{\sigma}_\omega} - \frac{\hat{\eta}_\omega}{\sigma_\omega} \right| + \left| \frac{\hat{\eta}_\omega}{\sigma_\omega} - \frac{\eta_\omega}{\sigma_\omega} \right| = |\hat{\eta}_\omega| \frac{1}{\hat{\sigma}_\omega} \frac{1}{\sigma_\omega} \frac{|\hat{\sigma}_\omega^2 - \sigma_\omega^2|}{\hat{\sigma}_\omega + \sigma_\omega} + \frac{1}{\sigma_\omega}|\hat{\eta}_\omega - \eta_\omega|$$

$$\leq \frac{C_\nu}{c \cdot \frac{c}{2} \cdot (c + \frac{c}{2})}|\hat{\sigma}_\omega^2 - \sigma_\omega^2| + \frac{1}{\sigma_\omega}|\hat{\eta}_\omega - \eta_\omega| = \frac{1}{c}|\hat{\eta}_\omega - \eta_\omega| + \frac{4\nu_z\nu^2}{3c^3}|\hat{\sigma}_\omega^2 - \sigma_\omega^2|. \tag{18}$$

The convergence rate of $\hat{\eta}_\omega$ and $\hat{\sigma}_\omega^2$ are proved in lemma 5 and lemma 6, respectively. Thus, with probability at least $1 - \delta$, the error is at most

$$\left(\frac{4}{c} + \frac{64\nu_z^2\nu^4}{3c^3}\right)\nu_z\nu\sqrt{\nu K}\ln\frac{4}{\delta} \cdot n^{-\frac{\beta-1}{2(\beta+p)}} + \left(\frac{2}{c} + \frac{8}{3c^3}\right)\nu_v\nu^2\sqrt{2\ln\frac{2}{\delta}} \cdot n^{-\frac{1}{2}} + 12\nu_z^2\nu^4 n^{-1}. \tag{19}$$

## D.1  The convergence rate of $\hat{\eta}_\omega$

**Lemma 5.** *Under Assumptions (Boundedness), (EVD) and (SRC), under the kernel parameters $\omega$, we have that with probability at least $1 - \delta$,*

$$|\hat{\eta}_\omega - \eta_\omega| \leq 2\nu_z\nu\left(2\ln(\frac{4}{\delta})\sqrt{\nu K}n^{-\frac{\beta-1}{2(\beta+p)}} + \nu\sqrt{2\ln\frac{2}{\delta}} \cdot n^{-\frac{1}{2}}\right). \tag{20}$$

*Proof.* The statistic $\eta_\omega$ includes both estimation error and random error. We use $\tilde{\eta}_\omega$ to denote the estimate of $\eta_\omega$ where the estimation error of the CMEs $\mu_{X|Z}$ and $\mu_{Y|Z}$ is removed on the finite samples $S = \{s_i\}_{i=1}^n$, which is

$$\tilde{\eta}_\omega = \frac{1}{n(n-1)}\sum_{(i,j)\in\mathbf{i}_2^n} h_{ij} = \frac{1}{n(n-1)}\sum_{(i,j)\in\mathbf{i}_2^n} k_{\mathcal{Z}}(z_i,z_j)r_{X|Z}(s_i,s_j)r_{Y|Z}(s_i,s_j), \tag{21}$$

where $r_{X|Z}(s_i, s_j) = \langle\phi_x(x_i) - \mu_{X|Z}(z_i), \phi_x(x_j) - \mu_{X|Z}(z_j)\rangle$. Then we can decompose $|\hat{\eta}_\omega - \eta_\omega|$ by

$$|\hat{\eta}_\omega - \eta_\omega| \leq |\hat{\eta}_\omega - \tilde{\eta}_\omega| + |\tilde{\eta}_\omega - \eta_\omega|, \tag{22}$$

where $|\hat{\eta}_\omega - \tilde{\eta}_\omega|$ represents the estimation error (bias), while $|\tilde{\eta}_\omega - \eta_\omega|$ is the random error (variance). It should be noted that the two factors contribute to the statistical estimation in a complex and mixed manner. We will separate them for a more intuitive analysis.

**Estimation bias.** The estimated bias is given by

$$|\hat{\eta}_\omega - \tilde{\eta}_\omega| = \frac{1}{n(n-1)} \sum_{i,j\neq i} |\hat{h}_{ij} - h_{ij}|, \tag{23}$$

with $|\hat{h}_{ij} - h_{ij}| = k_{\mathcal{Z}}(z_i, z_j)|\hat{r}_{X|Z}(x_i, x_j)\hat{r}_{Y|Z}(y_i, y_j) - r_{X|Z}(x_i, x_j)r_{Y|Z}(y_i, y_j)|$. To simplify the notation, we use $A_{ij}$ to replace $r_{X|Z}(s_i, s_j)$ with $\hat{A}_{ij} = \hat{\mu}_{X|Z}(x_i, x_j)$, and use $B_{ij}$ and $\hat{B}_{ij}$ to replace the corresponding $r_{Y|Z}(y_i, y_j)$ and $\hat{B}_{ij} = \hat{\mu}_{X|Z}(s_i, s_j)$, respectively. Then given $k_{\mathcal{Z}} \in (0, \nu_z]$, we can derive that

$$|\hat{h}_{ij} - h_{ij}| = k_{\mathcal{Z}}(z_i, z_j)|\hat{A}_{ij}\hat{B}_{ij} - A_{ij}B_{ij}| = k_{\mathcal{Z}}(z_i, z_j)|(\hat{A}_{ij} - A_{ij})\hat{B}_{ij} + A_{ij}(\hat{B}_{ij} - B_{ij})|. \tag{24}$$

where

$$\begin{aligned}
A_{ij} - \hat{A}_{ij} &= \langle\phi_x(x_i) - \mu_{X|Z}(z_i), \phi_x(x_j) - \mu_{X|Z}(z_j)\rangle - \langle\phi_x(x_i) - \hat{\mu}_{X|Z}(z_i), \phi_x(x_j) - \hat{\mu}_{X|Z}(z_j)\rangle \\
&= \langle\phi_x(x_i), \hat{\mu}_{X|Z}(z_j) - \mu_{X|Z}(z_j)\rangle + \langle\phi_x(x_j), \hat{\mu}_{X|Z}(z_i) - \mu_{X|Z}(z_i)\rangle + \\
&\quad \langle\mu_{X|Z}(z_i), \mu_{X|Z}(z_j)\rangle - \langle\hat{\mu}_{X|Z}(z_i), \hat{\mu}_{X|Z}(z_j)\rangle \\
&= \langle\phi_x(x_i), \hat{\mu}_{X|Z}(z_j) - \mu_{X|Z}(z_j)\rangle + \langle\phi_x(x_j), \hat{\mu}_{X|Z}(z_i) - \mu_{X|Z}(z_i)\rangle + \\
&\quad \langle\hat{\mu}_{X|Z}(z_i), \mu_{X|Z}(z_j) - \hat{\mu}_{X|Z}(z_j)\rangle + \langle\mu_{X|Z}(z_j), \mu_{X|Z}(z_i) - \hat{\mu}_{X|Z}(z_i)\rangle \\
&= \langle\phi_x(x_i) - \hat{\mu}_{X|Z}(z_i), \hat{\mu}_{X|Z}(z_j) - \mu_{X|Z}(z_j)\rangle + \langle\phi_x(x_j) - \mu_{X|Z}(z_j), \hat{\mu}_{X|Z}(z_i) - \mu_{X|Z}(z_i)\rangle.
\end{aligned} \tag{25}$$

According to the Cauchy-Schwarz inequality and the bound of CME in theorem 7, we can derive that with probability at least $1 - 4e^{-\tau}$ for $\tau \geq 1$,

$$|A_{ij} - \hat{A}_{ij}| \leq |\phi_x(x_i) - \hat{\mu}_{X|Z}(z_i)| \cdot |\hat{\mu}_{X|Z}(z_j) - \mu_{X|Z}(z_j)| + |\phi_x(x_j) - \mu_{X|Z}(z_j)| \cdot |\hat{\mu}_{X|Z}(z_i) - \mu_{X|Z}(z_i)|$$

$$\leq 2\sqrt{\nu} \cdot \sqrt{\tau^2 K n^{\frac{\beta-1}{\beta+p}}} = 2\tau\sqrt{\nu K} n^{-\frac{\beta-1}{2(\beta+p)}}. \tag{26}$$

The same notations $B_{ij}$ and $\hat{B}_{ij}$ are also applied to $Y$ with the same bound. Given the bound $k_{\mathcal{Z}} \in (0, \nu_z]$, $|A_{ij}| \leq \nu$, and $|\hat{A}_{ij}| \leq \nu$, we can derive that

$$\begin{aligned}
|\hat{h}_{ij} - h_{ij}| &= k_{\mathcal{Z}}(z_i, z_j)|(\hat{A}_{ij} - A_{ij})\hat{B}_{ij} + A_{ij}(\hat{B}_{ij} - B_{ij})| \\
&\leq k_{\mathcal{Z}}(z_i, z_j)\left[|(\hat{A}_{ij} - A_{ij})\hat{B}_{ij}| + |A_{ij}(\hat{B}_{ij} - B_{ij})|]\right] \\
&\leq 2\nu_z \cdot \nu \cdot 2\tau\sqrt{\nu K} n^{-\frac{\beta-1}{2(\beta+p)}} = 4\tau\nu_z\nu^{\frac{3}{2}}\sqrt{K} n^{-\frac{\beta-1}{2(\beta+p)}}.
\end{aligned} \tag{27}$$

Therefore, we can give the estimate bias bound

$$|\hat{\eta}_\omega - \tilde{\eta}_\omega| = \frac{1}{n(n-1)} \sum_{i,j\neq i} |\hat{h}_{ij} - h_{ij}| \leq 4\tau\nu_z\nu^{\frac{3}{2}}\sqrt{K} n^{-\frac{\beta-1}{2(\beta+p)}}. \tag{28}$$

**Random error.** For the random error, we can use McDiarmid's inequality to obtain the bound. We first consider replacing $s_k$ with $s_k' = (x_k', y_k', z_k')$ and keep the remaining samples in $S$ the same. Given $\hat{h}$ and $h$ are bounded in $[-C_\nu, C_\nu]$, the difference between $\hat{\eta}_\omega$ and the new substitution $\hat{\eta}_\omega'$ with $s_k'$ is given by

$$\begin{aligned}
|\tilde{\eta}_\omega - \tilde{\eta}_\omega'| &\leq \frac{1}{n(n-1)}\left(\sum_{i\neq k}|h_{ik'} - h_{ik}| + \sum_{j\neq k}|h_{k'j} - h_{kj}|\right) \\
&\leq \frac{2}{n(n-1)}\sum_{i\neq k}|h_{ik'} - h_{ik}| = \frac{4C_\nu}{n}.
\end{aligned} \tag{29}$$

Now using McDiarmid's inequality, for come certain and fixed $\omega$, with probality at least $1 - \delta$, the random error function $\Delta_\eta(\omega) := \hat{\eta}_\omega - \hat{\eta}'_\omega$:

$$|\hat{\eta}_\omega - \eta_\omega| \le 2\nu_z\nu^2\sqrt{\frac{2}{n}\ln\frac{2}{\delta}}. \tag{30}$$

Finally, by combining Eq. 28 and Eq. 30 with $\tau = \ln(4/\delta)$, we can finish the proof:

$$
\begin{aligned}
|\hat{\eta}_\omega - \eta_\omega| &\le 4\ln(\frac{4}{\delta})\nu_z\nu^{\frac{3}{2}}\sqrt{K}n^{-\frac{\beta-1}{2(\beta+p)}} + 2\nu_z\nu^2\sqrt{\frac{2}{n}\ln\frac{2}{\delta}} \\
&= 2\nu_z\nu\left(2\ln(\frac{4}{\delta})\sqrt{\nu K}n^{-\frac{\beta-1}{2(\beta+p)}} + \nu\sqrt{2\ln\frac{2}{\delta}}\cdot n^{-\frac{1}{2}}\right).
\end{aligned}
\tag{31}
$$

Since $\beta \in (1, 2]$ and $p \in (0, 1]$, $\frac{\beta-1}{2(\beta+p)} < \frac{1}{2}$ always holds. Therefore, even in the well-specified cases, the error is primarily dominated by the estimation bias with its slower convergence rate. As for the misspecified scenario with slower learning rate $O((\frac{\log n}{n})^{\beta-1})$ and $\beta \in (0, 1)$, the estimation error will become increasingly dominant.

### D.2 The convergence of $\hat{\sigma}_\omega$.

**Lemma 6.** *Under Assumptions (Boundedness), (EVD) and (SRC), under the kernel parameters $\omega = (\omega_x, \omega_y)$, we have that with probability at least $1 - \delta$,*

$$\left|\hat{\sigma}_\omega^2 - \sigma_\omega^2\right| \le 2\nu_z\nu^2\left(8\ln\frac{4}{\delta}\nu_z\nu\sqrt{\nu K}n^{-\frac{\beta-1}{2(\beta+p)}} + \sqrt{8\ln\frac{2}{\delta}}\cdot n^{-\frac{1}{2}} + 9\nu_z\nu^2 n^{-1}\right). \tag{32}$$

*Proof.* We analyze the convergence of $\sigma$ in the same way by decomposing it into estimation error and random error, which is

$$\left|\hat{\sigma}_\omega^2 - \sigma_\omega^2\right| \le \left|\hat{\sigma}_\omega^2 - \tilde{\sigma}_\omega^2\right| + \left|\tilde{\sigma}_\omega^2 - \sigma_\omega^2\right|, \tag{33}$$

**Estimation bias.** By denoting $A_i = \dfrac{\sum_{j\ne i}\hat{h}_{ij}}{n-1} - \hat{\eta}_\omega$ and $B_i = \dfrac{\sum_{j\ne i}h_{ij}}{n-1} - \tilde{\eta}_\omega$. The estimated bias of $\sigma_\omega^2$ is given by

$$
\begin{aligned}
\left|\hat{\sigma}_\omega^2 - \tilde{\sigma}_\omega^2\right| &= \frac{1}{n}\sum_i\left|\left(\frac{\sum_{j\ne i}\hat{h}_{ij}}{n-1} - \hat{\eta}_\omega\right)^2 - \left(\frac{\sum_{j\ne i}h_{ij}}{n-1} - \tilde{\eta}_\omega\right)^2\right| \\
&= \frac{1}{n}\sum_i\left|A_i^2 - B_i^2\right| \le \frac{1}{n}\sum_i|A_i + B_i|\cdot|A_i - B_i|| \\
&\le \frac{1}{n}\sum_i\left|\frac{\sum_{j\ne i}\hat{h}_{ij}}{n-1} - \hat{\eta}_\omega + \frac{\sum_{j\ne i}h_{ij}}{n-1} - \tilde{\eta}_\omega\right|\cdot\left|\frac{\sum_{j\ne i}\hat{h}_{ij}}{n-1} - \hat{\eta}_\omega - \frac{\sum_{j\ne i}h_{ij}}{n-1} + \tilde{\eta}_\omega\right| \\
&\le \frac{1}{n}\sum_i\left|\frac{\sum_{j\ne i}(\hat{h}_{ij} + h_{ij})}{n-1} - (\hat{\eta}_\omega + \tilde{\eta}_\omega)\right|\cdot\left(\frac{\sum_{j\ne i}|\hat{h}_{ij} - h_{ij}|}{n-1} + |\hat{\eta}_\omega - \tilde{\eta}_\omega|\right) \\
&\le \frac{1}{n}\sum_i\left|\frac{\sum_{j\ne i}(\hat{h}_{ij} + h_{ij})}{n-1} - (\hat{\eta}_\omega + \tilde{\eta}_\omega)\right|\cdot 2|\hat{h}_{ij} - h_{ij}| \\
&= 2\epsilon_{\hat{\eta}}\cdot\frac{1}{n}\sum_i\left|\frac{\sum_{j\ne i}(\hat{h}_{ij} + h_{ij})}{n-1} - \frac{\sum_{k,l\ne k}(\hat{h}_{kl} + h_{kl})}{n(n-1)}\right|,
\end{aligned}
\tag{34}
$$

where $\epsilon_{\hat{\eta}} = |\hat{h}_{ij} - h_{ij}|$ is derived in eq. (27). Since $\sum_{k,l\neq k}(\hat{h}_{kl} + h_{kl}) = \sum_{k\neq i}\sum_{l\neq k}(\hat{h}_{kl} + h_{kl}) + \sum_{l\neq i}(\hat{h}_{il} + h_{il})$. We can derive that

$$
\begin{aligned}
\left|\hat{\sigma}_\omega^2 - \tilde{\sigma}_\omega^2\right| &\leq 2\epsilon_{\hat{\eta}} \cdot \frac{1}{n}\sum_i \left| \frac{n\sum_{j\neq i}(\hat{h}_{ij} + h_{ij})}{n(n-1)} - \frac{\sum_{l\neq i}(\hat{h}_{il} + h_{il})}{n(n-1)} - \frac{\sum_{k\neq i}\sum_{l\neq k}(\hat{h}_{kl} + h_{kl})}{n(n-1)} \right| \\
&= 2\epsilon_{\hat{\eta}} \cdot \frac{1}{n}\sum_i \left| \frac{(n-1)\sum_{j\neq i}(\hat{h}_{ij} + h_{ij})}{n(n-1)} - \frac{\sum_{k\neq i}\sum_{l\neq k}(\hat{h}_{kl} + h_{kl})}{n(n-1)} \right| \\
&\leq 2\epsilon_{\hat{\eta}} \cdot \frac{1}{n}\sum_i \left| \frac{\sum_{j\neq i}(\hat{h}_{ij} + h_{ij})}{n} \right| + 2\epsilon_{\hat{\eta}} \cdot \frac{1}{n}\sum_i \left| \frac{\sum_{k\neq i}\sum_{l\neq k}(\hat{h}_{kl} + h_{kl})}{n(n-1)} \right| \\
&\leq 2\epsilon_{\hat{\eta}} \cdot \frac{1}{n} \cdot n \frac{(n-1)\cdot 2C_\nu}{n} + 2\epsilon_{\hat{\eta}} \cdot \frac{1}{n} \frac{n(n-1)\cdot 2C_\nu}{n(n-1)} \\
&= 4C_\nu \epsilon_{\hat{\eta}} = 16\tau\nu_z^2\nu^3\sqrt{\nu K} n^{-\frac{\beta-1}{2(\beta+p)}}.
\end{aligned}
$$

(35)

**Random Error.** We again use McDiarmid's inequality to obtain the bound for the random error of the asymptotic variance which is $\Delta_\sigma(\omega) = |\tilde{\sigma}_\omega^2 - \sigma_\omega^2|$, where

$$\sigma_\omega^2 = \mathbb{E}_i[\mathbb{E}_j[h_{ij}] - \eta_\omega]^2. \tag{36}$$

and the estimate without the estiamate bias on finite sample set S, which is

$$\tilde{\sigma}_\omega^2 = \frac{1}{n}\sum_i \left[ \frac{\sum_{j\neq i} h_{ij}}{n-1} - \tilde{\eta}_\omega \right]^2. \tag{37}$$

We again consider replacing $s_k$ with $s'_k$ and keep the remaining samples the same with the counterpart notations $\tilde{\sigma}'_\omega$ and $\tilde{\eta}_\omega$. Given that $|h_{ij}| \leq C_\nu$, $|\tilde{\eta}_\omega| \leq C_\nu$, $|h_{ik} - h'_{ik}| \leq 2C_\nu$ and $|\hat{\eta}_\omega - \hat{\eta}'_\omega| \leq \frac{4C_\nu}{n}$ proved in eq. (29), we can derive that

$$
\begin{aligned}
\left|\tilde{\sigma}_\omega^2 - \tilde{\sigma}_\omega'^2\right| &= \left| \frac{1}{n}\sum_i \left( \frac{\sum_{j\neq i} h_{ij}}{n-1} - \tilde{\eta}_\omega \right)^2 - \frac{1}{n}\sum_i \left( \frac{\sum_{j\neq i} h'_{ij}}{n-1} - \tilde{\eta}'_\omega \right)^2 \right| \\
&\leq \frac{1}{n}\sum_i \left| \frac{\sum_{j\neq i}(h_{ij} + h'_{ij})}{n-1} - (\tilde{\eta}_\omega + \tilde{\eta}'_\omega) \right| \cdot \left( \frac{\sum_{j\neq i}|h_{ij} - h'_{ij}|}{n-1} + |\tilde{\eta}_\omega - \tilde{\eta}'_\omega| \right) \\
&\leq \frac{1}{n}\sum_{i\neq k} \left| \frac{\sum_{j\neq i}(h_{ij} + h'_{ij})}{n-1} - (\tilde{\eta}_\omega + \tilde{\eta}'_\omega) \right| \cdot \left( \frac{|h_{ik} - h'_{ik}|}{n-1} + |\tilde{\eta}_\omega - \tilde{\eta}'_\omega| \right) + \\
&\quad \frac{1}{n} \left| \frac{\sum_{j\neq k}(h_{kj} + h'_{kj})}{n-1} - (\tilde{\eta}_\omega + \tilde{\eta}'_\omega) \right| \cdot \left( \frac{\sum_{j\neq k}|h_{kj} - h'_{kj}|}{n-1} + |\tilde{\eta}_\omega - \tilde{\eta}'_\omega| \right) \\
&\leq \frac{1}{n} \cdot (n-1)\left( \frac{(n-1)\cdot 2C_\nu}{n-1} + 2C_\nu \right) \cdot \left( \frac{2C_\nu}{n-1} + \frac{4C_\nu}{n} \right) \\
&\quad + \frac{1}{n}\left( \frac{(n-1)\cdot 2C_\nu}{n-1} + 2C_\nu \right) \cdot \left( \frac{(n-1)\cdot 2C_\nu}{n-1} + \frac{4C_\nu}{n} \right) \\
&= \frac{1}{n} \cdot 4C_\nu \cdot \left( 2C_\nu + 4C_\nu \frac{n-1}{n} + 2C_\nu + \frac{4C_\nu}{n} \right) = \frac{8C_\nu}{n}.
\end{aligned}
$$

(38)

Simply applying McDiamid's to $\tilde{\sigma}_\omega^2$, we obtain that with probability at least $1 - \delta$,

$$\left|\tilde{\sigma}_\omega^2 - \mathbb{E}[\tilde{\sigma}_\omega^2]\right| \leq 4C_\nu\sqrt{\frac{2}{n}\ln\frac{2}{\delta}}. \tag{39}$$

Now we consider the bound $|\mathbb{E}[\tilde{\sigma}_\omega^2] - \sigma_\omega^2|$. By the definition, we can rewrite $\sigma_\omega^2$ and $\mathbb{E}[\tilde{\sigma}_\omega^2]$ with

$$
\begin{aligned}
\mathbb{E}[\tilde{\sigma}_\omega^2] &= \frac{1}{n^3}\sum_{ilk}\mathbb{E}[h_{il}h_{ik}] - \frac{1}{n^4}\sum_{ijlk}\mathbb{E}[h_{ij}h_{kl}] \quad \text{and} \\
\sigma_\omega^2 &= \frac{1}{n^3}\sum_{ilk}\mathbb{E}[h_{12}h_{13}] - \frac{1}{n^4}\sum_{ijlk}\mathbb{E}[h_{12}h_{34}],
\end{aligned}
$$

(40)

where the number subscripts in $\sigma_\omega^2$ corresponding to $i$, $j$, $k$, and $l$ are mutually distinct and $E[h_{12}h_{34}] < C_\nu^2$, $E[h_{ij}h_{kl}] \le C_\nu^2$.

For the first term, we obtain

$$\left| \frac{1}{n^3} \sum_{ilk} \mathbb{E}[h_{il}h_{ik}] - \frac{1}{n^3} \sum_{ilk} \mathbb{E}[h_{12}h_{13}] \right| < \left( 1 - \frac{(n)_3}{n^3} \right) \cdot |C_\nu^2 + C_\nu^2| = \left( \frac{3}{n} - \frac{2}{n^2} \right) \cdot 2C_\nu^2. \quad (41)$$

The second term can be handled similarly:

$$\left| \frac{1}{n^4} \sum_{ijlk} \mathbb{E}[h_{ij}h_{kl}] - \frac{1}{n^4} \sum_{ijlk} \mathbb{E}[h_{12}h_{34}] \right| < \frac{n^4 - n(n-1)(n-2)(n-3)}{n^4} \cdot 2C_\nu^2 = \left( \frac{6}{n} - \frac{11}{n^2} + \frac{6}{n^3} \right) \cdot 2C_\nu^2. \quad (42)$$

Therefore, since $13/n^2 > 6n^3$ for $n \ge 1$, we have

$$|\mathbb{E}[\tilde{\sigma}_\omega^2] - \sigma_\omega^2| < 2C_\nu^2 \left( \frac{9}{n} - \frac{13}{n^2} + \frac{6}{n^3} \right) < \frac{18C_\nu^2}{n}. \quad (43)$$

Combining eq. (39) and eq. (43), we can give the random error bound under the residuals can be well estimated, which is

$$|\tilde{\sigma}_\omega^2 - \sigma_\omega^2| \le |\tilde{\sigma}_\omega^2 - \mathbb{E}[\tilde{\sigma}_\omega^2]| + |\mathbb{E}[\tilde{\sigma}_\omega^2] - \sigma_\omega^2| = 4C_\nu \sqrt{\frac{2}{n} \ln \frac{2}{\delta}} + \frac{18C_\nu^2}{n}. \quad (44)$$

And finally, we combine eq. (44) and eq. (35) to get the bound of estimate asymptotic variance which is

$$\begin{aligned}
\left| \hat{\sigma}_\omega^2 - \sigma_\omega^2 \right| &\le 16\tau\nu_z^2\nu^3\sqrt{\nu K} n^{-\frac{\beta-1}{2(\beta+p)}} + 4C_\nu \sqrt{\frac{2}{n} \ln \frac{2}{\delta}} + \frac{18C_\nu^2}{n} \\
&= 2\nu_z\nu^2 \left( 8\ln\frac{4}{\delta}\nu_z\nu\sqrt{\nu K} n^{-\frac{\beta-1}{2(\beta+p)}} + \sqrt{8\ln\frac{2}{\delta}} \cdot n^{-\frac{1}{2}} + 9C_\nu n^{-1} \right).
\end{aligned} \quad (45)$$

Recall that $c$ is a small positive constant, since $\sigma_\omega^2 \ge c^2$ on $\bar{\Omega}_c$, according to eq. (45), when the sample size $n_0$ holds:

$$2\nu_z\nu^2 \left( 8\ln\frac{4}{\delta}\nu_z\nu\sqrt{\nu K} n_0^{-\frac{\beta-1}{2(\beta+p)}} + \sqrt{8\ln\frac{2}{\delta}} \cdot n_0^{-\frac{1}{2}} + 9C_\nu n_0^{-1} \right) \le \frac{c}{2}, \quad (46)$$

we have $\hat{\sigma}_\omega \ge \frac{c}{2}$ with at least probalibity $1 - \delta/2$.

## D.3  CME convergence.

The analysis of the learning rate in vector-valued regression depends on standard assumptions such as (EVD) and (SRC), which impose constraints on the input space, interpolation operators, and the smoothness of the target regression. These assumptions help to ensure that the model's convergence behavior is well-defined and that the learning rate reflects the complexity of the regression task, influenced by factors such as the smoothness of the target function and the properties of the interpolation operators.

In SRC, the eigenvalues associated with functions in $[\text{HS}]^\beta$ decay at a rate that is influenced by the parameter $\beta$. Specifically, when $\beta > 1$, $[\text{HS}]^\beta$ is embedded within HS, and the eigenvalues decay faster than those in HS, reflecting increased smoothness of the functions in $[\text{HS}]^\beta$. Conversely, when $0 < \beta < 1$, $[\text{HS}]^\beta$ includes HS as a subset, and the eigenvalues decay more slowly, indicating a larger function space compared to HS. Here, we follow Pogodin et al. [2024] and focus on well-specified cases with $\beta \in (1, 2]$. For the misspecified case where $\beta \in (0, 1)$, the corresponding convergence rates from [Li et al., 2022, Theorem 2] can be directly applied and will not be discussed in the paper.

**Theorem 7.** *Li et al. [2022] Under the Assumption EVD, SRC and Boundedness, for the well-specified cases where $\gamma = 1$, $\alpha = 1$, $\beta \in (1, 2]$, there is a constant $K > 0$ independent of $n \ge 1$ and $\tau \ge 1$, then*

$$\|\hat{\mu}_{X|Z} - \mu_{X|Z}\|^2 \le \tau^2 K n^{-\frac{\beta-1}{\beta+p}}, \quad (47)$$

*is satisfied for sufficiently large $n$ with $P^n$-probability not less than $1 - 4e^{-\tau}$.*

# E    Implementation and Baseline Details

We present the implementation details of both our proposed method and the synthetic dataset for conditional independence test and the causal discovery tasks.

## E.1    Model hyperparameter settings

Our method's parameters mainly exist in the kernel ridge regression, the process of learning the parameters in $\phi_z$ and the final testing procedure. (1) For the kernel ridge regression, there are three trainable parameters, the amplitude $A$, the bandwidth involved in $K_Z^R$, denoted as $\sigma_z^r$, and the regularization parameter $\varepsilon$. To ensure stability during the training process, we have constrained their value ranges, and the amplitude $A$ is limited to the range of $[10^{-3}, 10^3]$. The bandwidth $\sigma_z^r$ is a vector whose dimensions are the same as those of the conditioning variable $Z$, with values constrained to $[10^{-2}, 10^2]$. The regularization parameter $\varepsilon$ is constrained to $[10^{-10}, 1]$. We use marginal likelihood as the loss function and the L-BFGS-B algorithm [Liu and Nocedal, 1989] to optimize and update these parameters. (2) In the final test stage, we use the weighted sum of $\chi^2$ approximation to simulate the null distribution. Following the default setting in [Zhang et al., 2011], we drop all $\tilde{\lambda}_k$ which are smaller than $10^{-5}$ for computational efficiency. We sampled a total of 5000 $\check{T}_b$ values according to Eq. 15, and obtained the $p$-value which is the rate that $\check{T}_b > \hat{C}_{\text{KCI}_b}^2$.

## E.2    Synthetic Data Generation Settings

**Implementation details of Synthetic CI dataset.** In the CI testing task, we assume $X$ and $Y$ are the dependent variable of $Z$. To examine Type I errors, $X$ and $Y$ were generated according to the following nonlinear additive function model:

$$X = f(W^\top Z) + E, \tag{48}$$

where $W \sim \mathcal{N}(0, I_{d_Z})$ and $d_Z$ represents the dimension of $Z$, $f$ was randomly chosen from the *linear*, *sin*, *cos*, $x^2$, $2^x$ and $\exp(x)$ with the same probability of being selected. When $f$ is neither *sin* nor *cos*, we multiply $W^\top Z$ by an additional $1/\sqrt{d_Z}$ to balance the scale of the function and noise. For *linear* case, the data generation process follows $X/Y = Z/\sqrt{d_Z} + E$. The noise term $E$ was randomly generated from either a normal distribution $\mathcal{N}(0, 1)$ or a uniform distribution $U(-1, 1)$ with equal probability. To test Type II error, we add the same latent variable $T$ to both $X$ and $Y$ with $T \sim \mathcal{N}(0, 1)$. Then the dependent variable, e.g. X, is generated as follows:

$$X = f(W^\top Z) + E + \alpha T, \tag{49}$$

$Y$ follows the same generating process with the same variable $T$.

**Implementation details of graph dataset for causal discovery.** In the synthetic graph data for causal discovery task, we generate cases with varying graph densities. The graph density is measured by the ratio of the number of edges to the maximum possible number of edges in the graph; a smaller graph density indicates fewer edges in the graph, while a larger density indicates a denser graph. For each graph density, we generate 10 cases where the variables and relationships are randomly generated. For each cases involves 10 variables with sample sizes of $n = 500$, which are evenly divided into training data and testing data. For each variable $X_i$ in the graph, the data was generated according to

$$X_i = f_i(W_i^\top PA_i) + E_i, \tag{50}$$

where $PA_i$ represents the parent nodes of $X_i$ in the graph and the weight matrix $W_i \sim \mathcal{N}(0, I)$. $f_i$ is equally likely to be sampled from *linear, sin, cos, exp* and $x^2$. Each function class in $f_i$ all has the same probability of being selected, and within the equal probability of each parameters setting. If one of the variables has no parent nodes in the graph, it follows a standard normal distribution. $E_i$ represents the noise variable, randomly following either a Gaussian distribution $\mathcal{N}(0, 1)$ or a uniform distribution $U(-1, 1)$ with equal probability. For each graph density, we generated 20 realizations. We set the significance level of $\alpha = 0.05$. All the data is similarly divided into training and testing sets with the same number of samples. We also use the weighted sum of $\chi^2$ approximation to simulate the null distribution for testing on the testing data.

### E.3    Configuration details for runtime benchmarking.

In Section 5.1.3, we use the default settings as described in 5: each kernel has 10 candidate parameter values, resulting in a total of $10 \times 10$ parameter combinations. Consequently, Power involves 20 CME computations, whereas Median only requires 2. We leverage the *joblib* package, specifically *Parallel* and *delayed*, to parallelize the learning process for the 20 CMEs in Power and the 2 CMEs in Median. For each sample size, we randomly generate 50 cases. All experiments were conducted on an Intel 14700K CPU platform with 32GB of RAM, without GPU acceleration.

### E.4    Conditional independence testing baselines

In this section, we provide a brief overview of the baseline methods used in the main text, along with their parameter settings.

**CIRCE** (Conditional Independence Regression CovariancE, [Pogodin et al., 2022]) is a simplified version of KCI, which only considers the correlation between $\phi_y(Y)$ and the regression residuals of $Z$ to $\phi_{xz}(X, Z)$, i.e. $\phi_{\ddot{x}|z}(X, Z) = \phi_{\ddot{x}}(X, Z) - \mathbb{E}[\phi_{\ddot{x}}(X, Z) \mid Z]$ with $\ddot{X} = (X, Z)$. As explained in Theorem 2, any function $g(X, Z) \in L^2$ can capture the general relationship between $X$ and $Z$. Utilizing the reproducing property, the residual feature map $\phi_{\ddot{x}|z}(X, Z)$ effectively eliminates the influence of $Z$ on $X$. Intuitively, this residual $\phi_{\ddot{x}|z}(X, Z)$ thus represents the component of $X$ that cannot be explained by $Z$. Thus, if $\phi_{\ddot{x}|z}(X, Z)$ is independent of $Y$, then we can conclude that $X$ and $Y$ are conditionally independent given $Z$. Formally, CIRCE has the following form:
$$T_{\text{CIRCE}} = \mathbb{E}[\phi_z(Z) \otimes \phi_y(Y) \otimes (\phi_x(X) - \mu_{X|Z}(Z))]. \tag{51}$$
Correspondingly, CIRCE also has an MMD-like biased estimator:
$$\hat{T}_{\text{CIRCE}} = \frac{1}{n(n-1)} \text{Tr}(HK_Z H(K_Y \odot R_{X|Z})). \tag{52}$$
and can similarly use weighted (infinite) sum of $\chi^2$ variables or Gamma approximation to estimate the null distribution for conducting CI testing.

In CIRCE, we follow the original settings of CIRCE: we use the median heuristic to initialize the parameters of $\phi_z$, $\phi_y$ and $\phi_x$. We also use a Gaussian process to estimate the conditional mean embedding $\mu_{X|Z}$, with parameters set identical to those used in our Power method.

**GCM** (Generalised Covariance Measure, [Shah and Peters, 2020]) considers the relation in $L^2$ space. For one sample pair $(x_i, y_i, z_i)$, it considers the product between residuals from the regression:
$$R_i = [x_i - \hat{f}(z_i)] \cdot [y_i - \hat{g}(z_i)],$$
where $\hat{f}$ is the estimates of the conditional mean $f(Z) = \mathbb{E}_Z[X \mid Z]$, and $\hat{g}$ is defined as the same. Then it defines the statistic with $n$ observations:
$$T = \frac{\sqrt{n} \sum_{i=1}^n R_i}{\sqrt{\left(\frac{1}{n} \sum_{i=1}^n (R_i)^2 - \left(\frac{1}{n} \sum_{i=1}^n R_i\right)^2\right)}}.$$
And it $p$-value is computed as $p = 2(1 - \Phi(|T|))$ for the standard normal CDF $\Phi$. We use the default regression model and parameter settings from GCM [3]: The conditional mean is estimated using cross-validated linear Lasso with 5 folds. the significance level is set as $0.05$ which is consistent with other approaches.

**RBPT2** (The Rao-Blackwellized Predictor Test, [Polo et al., 2023]) involve a regression chain: it first needs to estimate $g(Y, Z) = [X \mid Y, Z]$. Then with the trained $g(Y, Z)$, it estimates $h(Z) = [g(Y, Z) \mid Z]$. The statistic is defined to compare the difference between their predicted results and the residuals of the real value of $X$, which is
$$T_i = l(h(z_i), x_i) - l(g(y_i, z_i), x_i), \quad S = \frac{\sqrt{n} \sum_{i=1}^n T_i}{\sqrt{\left(\frac{1}{n} \sum_{i=1}^n (T_i)^2 - \left(\frac{1}{n} \sum_{i=1}^n T_i\right)^2\right)}},$$
where $l$ is MSE loss $l = (g - x)^2$ and its $p$-value The p-value is then computed as $p = 1 - \Phi(S)$. We follow its default model and parameter setting[4].

---

[3] https://github.com/LaplaceSansouci/GeneralizedCovarianceMeasure
[4] https://github.com/felipemaiapolo/cit

## F  More Experimental results and discussion

### F.1  Real data.

#### F.1.1  Car insurance data.

Following the setup of [Polo et al., 2023], we test our methods on the car insurance dataset for conditional independence testing. The car insurance data[5] encompasses four US states (California, Illinois, Missouri and Texas) and includes information from numerous insurance providers compiled at the ZIP code granularity. The data offers a risk metric and the insurance price levied on a hypothetical customer with consistent attributes from every ZIP code. ZIP codes are categorized as either minority or non-minority, contingent on the percentage of non-white residents. The variables in consideration are $Z$, denoting the driving risk; $X$, an indicator for minority ZIP codes; and the insurance price $Y$. A pertinent question revolves around the validity of the null hypothesis $H_0 : X \perp\!\!\!\perp Y \mid Z$, essentially questioning if demographic biases influence pricing.

Since this is a real dataset, the full distribution and the true CI relationship between $X$ and $Y$ given $Z$ are unknown. Therefore, following [Polo et al., 2023], we discretize the conditioning variable $Z$ into 20 bins and shuffle the $Y$ values corresponding to each discrete $Z$ value. If a test maintains Type-I error control, we expect it to reject $H_0$ for at most $\alpha = 0.05$ of the companies in each state. In the second part, we use the unshuffled data for CI testing and focus on assessing the power of our methods. Following the default setting in Polo et al. [2023], the dataset is split $70/30\%$ for training and testing. We conducted a total of 5 experiments, each time randomly selecting 10 seeds, and reported the average Type-I error rate and the average $p$-value. This experiment is to assess whether CI methods can effectively control Type I error on the shuffled dataset, while determining on the unshuffled dataset whether the state exists where demographic biases influence pricing.

Figure 3a illustrates that both the Median and Power methods are generally effective in controlling Type I error, with Power performing slightly better than Median. Specifically, the Type I error for the Median method in Missouri exceeds $\alpha = 0.05$, whereas Power maintains a strict control at 0.05. Figure 3b presents the test results on the original unshuffled data. All methods show relatively low p-values, leading to the conclusion that all states likely exhibit varying degrees of discrimination against minorities in ZIP codes. The severity, in descending order, is Illinois, Texas, Missouri, and California. This result is consistent with the findings from [Angwin et al., 2022], indicating that our method is capable of correctly identifying CI relationships in the real world.

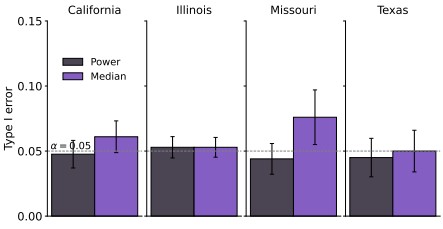    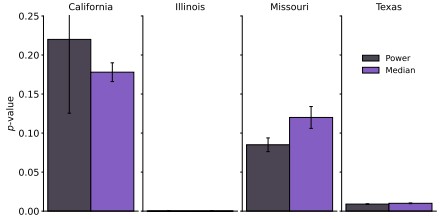

(a) Performance on *shuffled* data.  (b) Performance on *unshuffled* data.

Figure 3: Performance comparison on car insurance dataset.

#### F.1.2  Causal Discovery Benchmarks.

In Section 5.2, we apply our method to two widely-used real-world causal discovery benchmarks, SACHs [Sachs et al., 2005] and CHILD [Spiegelhalter et al., 1993] datasets. SACHs dataset contains single-cell measurements of protein and phospholipid components in human immune system signaling pathways, originally collected to study causal protein interactions under different experimental conditions. Its network comprises 11 variables and 17 edges. CHILD network consists of 20 variables with 25 edges, and its graph is a medium-sized causal graph that models congenital heart disease in newborns. The ground-truth graph consists of 20 nodes and 25 edges. Some example variables in the graph include Birth Asphyxia, Lung Flow, and Chest X-ray.

---

[5]Data description link

Following the default setting, the variables were extracted and preprocessed. In both benchmarks, the variables are discrete and take values ranging from 1 to 6. We evaluated the performance under different sample sizes. For each sample size, data were randomly sampled and the experiments were repeated 10 times. F1 score is a weighted average of precision and recall, calculated as

$$\text{F1} = \frac{2 \cdot \text{recall} \cdot \text{precision}}{\text{recall} + \text{precision}}. \tag{53}$$

. Table 2 reports the F1 score results for sample size $N = 500$. Additional results for various sample sizes are presented in Table 3 and 4. Across all sample sizes, Power with selected kernels consistently outperforms Median, demonstrating the effectiveness and applicability of our method in real-world relational networks.

Table 3: F1 score on SACHs with varying sample sizes $N$.

| F1 Score | $N = 200$ | 400 | 600 | 800 | 1000 |
|---|---|---|---|---|---|
| Median | $0.416 \pm 0.063$ | $0.550 \pm 0.047$ | $0.593 \pm 0.053$ | $0.648 \pm 0.032$ | $0.710 \pm 0.035$ |
| Power | $\mathbf{0.445 \pm 0.062}$ | $\mathbf{0.590 \pm 0.058}$ | $\mathbf{0.652 \pm 0.060}$ | $\mathbf{0.732 \pm 0.037}$ | $\mathbf{0.794 \pm 0.027}$ |

Table 4: F1 score on CHILD across $N$.

| F1 Score | $N = 200$ | 400 | 600 | 800 | 1000 |
|---|---|---|---|---|---|
| Median | $0.638 \pm 0.056$ | $0.723 \pm 0.063$ | $0.806 \pm 0.045$ | $0.832 \pm 0.021$ | $0.861 \pm 0.022$ |
| Power | $\mathbf{0.673 \pm 0.069}$ | $\mathbf{0.779 \pm 0.038}$ | $\mathbf{0.835 \pm 0.030}$ | $\mathbf{0.848 \pm 0.022}$ | $\mathbf{0.870 \pm 0.018}$ |

### F.2 More experiment result on synthetic data

In this section, we present experiments with additional baseline methods and on data with non-additive relationships to better assess the effectiveness of the KCI method under general data relationships and to evaluate the improvements brought by our approach over the median heuristic-based KCI (i.e. Median).

**Baselines.** We additionally include three conditional independence testing methods as baselines, namely KNN [Li et al., 2023], CCIT [Sen et al., 2017] and AKE [Scetbon et al., 2022].

- **KNN** [Li et al., 2023] proposes a permutation-based conditional independence test that leverages local K-nearest-neighbor sampling to approximate the conditional distribution and construct the null distribution for statistical inference. We used their default parameter settings, adopt XGBoost as the classifier, and set the number of repetitions $B = 200$ and the neighbor order $k = 7$.

- **CCIT** [Sen et al., 2017] proposes a model-powered, permutation-based conditional independence test, which uses neural networks to learn a representation of the conditional distribution and trains a classifier to distinguish between samples from the joint and null (permuted) distributions. We followed the default hyperparameter settings: we use 30 bootstrap iterations, and tuned the XGBoost classifier over the suggested grid of tree depths $\{6, 10, 13\}$, number of boosting rounds $\{100, 200, 300\}$, and column subsampling ratio fixed at 0.8.

- **AKE** [Scetbon et al., 2022] AKE is an asymptotic conditional independence test that estimates kernel-based covariance operators and derives the null distribution in closed form using operator-theoretic tools and analytic kernels. While it shares similarities with KCIT in relying on kernel embeddings and asymptotic theory, it differs by explicitly exploiting the structure of analytic kernels to achieve improved computational efficiency and theoretical interpretability. We followed the default settings recommended in the original AKE implementation, using Gaussian processes as regressors. All kernel functions involved are Gaussian kernels, with bandwidth parameters selected via the median heuristic.

**Nonlinear additive data.** We first conduct experiments on the original synthetic data with nonlinear additive relationships (Eq. 48), generated in the same way as described in Appendix E.2. All methods are repeated 500 times for each setting.

The experimental results are presented in Table 5. The last row of the table reports the testing time required by each method for a single case: we evaluate on samples with $d_Z = 5$, and compute the average testing time over 100 repetitions for each method and error type. As shown, both KNN and AKE effectively control the Type I error at the given significance level, whereas CCIT fails to do so under this setting. In terms of Type II error, AKE outperforms KNN. Overall, KCI (i.e., Median) and its kernel-selected version, Power, achieve the best performance.

Table 5: Type I and Type II Errors across different $d_Z$ settings.

|  | Method | $d_Z = 1$ | 3 | 5 | 7 | 9 | Testing time (s) |
|---|---|---|---|---|---|---|---|
| **Type I Error** | AKE | 0.06 | 0.04 | 0.04 | 0.06 | 0.04 | $3.463 \pm 0.030$ |
|  | KNN | 0.06 | 0.04 | 0.04 | 0.04 | 0.05 | $6.261 \pm 0.071$ |
|  | CCIT | 0.41 | 0.31 | 0.31 | 0.29 | 0.21 | $5.106 \pm 0.133$ |
|  | Median | 0.04 | 0.03 | 0.05 | 0.04 | 0.03 | $1.270 \pm 0.061$ |
|  | Power | 0.05 | 0.04 | 0.05 | 0.03 | 0.02 | $2.202 \pm 0.286$ |
| **Type II Error** | AKE | 0.15 | 0.31 | 0.44 | 0.57 | 0.68 | $3.164 \pm 0.042$ |
|  | KNN | 0.90 | 0.82 | 0.83 | 0.88 | 0.89 | $5.923 \pm 0.101$ |
|  | CCIT | 0.32 | 0.25 | 0.48 | 0.73 | 0.67 | $5.436 \pm 0.221$ |
|  | Median | 0.19 | 0.32 | 0.41 | 0.46 | 0.57 | $1.361 \pm 0.062$ |
|  | Power | 0.13 | 0.25 | 0.31 | 0.34 | 0.46 | $2.437 \pm 0.399$ |

**Non-additive data.** We further conduct experiments on a synthetic dataset generated under a non-additive functional relationship, defined as follows:

$$X = f(W^\top Z + E), \tag{54}$$

where the weight matrix $W$, function $f$, and noise term $E$ follow the same type and distribution as those in Eq. 48. All parameter settings remain the same as before, and each setting is evaluated over 500 repeated trials.

The results are shown in Table 6. The methods GCM, RBPT2, and CIRCE have been introduced in Appendix E.4, and their parameter settings remain unchanged. From the results, CCIT fails to effectively control the Type I error under this setting, while RBPT2 slightly exceeds the significance level. The other methods maintain Type I error rates close to the nominal level $\alpha = 0.05$. In terms of test power, the best performance is again achieved by the KCI-based methods, Median and Power, with Power further reducing the Type II error compared to Median.

## F.3 High-dimensional data

We evaluate the performance of our method and the baseline methods when the conditioning set $Z$ has high dimensionality to assess their statistical limits and identify potential performance bottlenecks under challenging settings. It should be noted that the high-dimensional setting presents a challenging scenario for CI tasks, as it introduces estimation bias that is difficult to mitigate, making it hard for models to learn accurate predictors or regressors [Shah and Peters, 2020]. We include all the baselines introduced earlier for comparison, including GCM, RBPT2, and CIRCE (introduced in Appendix E.4), as well as the additional methods KNN, CCIT, and AKE, described in Appendix F.2.

**Setting.** We conduct experiments on the nonlinear additive synthetic data, according to Eq. 48, with parameter distributions remaining the same as in the default setting. The key difference is that we now the performance is evaluated under higher-dimensional conditioning variables, with $d_Z = 10, 30,$ and 50.

**Result.** In Table 7, we provide experimental results on nonlinear additive synthetic data under high-dimensional conditioning setting. When the dimensionality increases, all comparison methods either fail to effectively control the Type I error (e.g., RBPT2, CIRCE, CCIT) or exhibit little to no

Table 6: Type I and Type II Errors across different $d_Z$ settings with more baseline methods.

| | Method | $d_Z = 1$ | 3 | 5 | 7 | 9 | Testing time (s) |
|---|---|---|---|---|---|---|---|
| **Type I Error** | AKE | 0.06 | 0.05 | 0.03 | 0.03 | 0.07 | 3.050 ± 0.586 |
| | KNN | 0.01 | 0.04 | 0.03 | 0.07 | 0.00 | 6.612 ± 0.026 |
| | CCIT | 0.39 | 0.34 | 0.38 | 0.25 | 0.17 | 5.314 ± 0.010 |
| | GCM | 0.06 | 0.05 | 0.05 | 0.07 | 0.06 | 1.213 ± 3.126 |
| | RBPT2 | 0.02 | 0.09 | 0.10 | 0.15 | 0.10 | 0.405 ± 0.031 |
| | CIRCE | 0.05 | 0.02 | 0.03 | 0.00 | 0.02 | 1.141 ± 0.301 |
| | Median | 0.04 | 0.05 | 0.05 | 0.06 | 0.06 | 1.436 ± 0.359 |
| | Power | 0.03 | 0.04 | 0.04 | 0.04 | 0.05 | 2.443 ± 0.496 |
| **Type II Error** | AKE | 0.19 | 0.60 | 0.77 | 0.70 | 0.71 | 2.900 ± 0.551 |
| | KNN | 0.82 | 0.76 | 0.84 | 0.81 | 0.82 | 6.603 ± 0.037 |
| | CCIT | 0.15 | 0.36 | 0.37 | 0.46 | 0.57 | 5.252 ± 0.006 |
| | GCM | 0.05 | 0.21 | 0.48 | 0.45 | 0.31 | 1.371 ± 3.032 |
| | RBPT2 | 0.56 | 0.64 | 0.72 | 0.80 | 0.81 | 0.265 ± 0.045 |
| | CIRCE | 0.21 | 0.52 | 0.74 | 0.66 | 0.68 | 1.005 ± 0.320 |
| | Median | 0.13 | 0.38 | 0.62 | 0.59 | 0.54 | 1.206 ± 0.231 |
| | Power | 0.07 | 0.32 | 0.53 | 0.50 | 0.37 | 2.461 ± 0.394 |

power against any alternative (e.g., GCM, AKE, KNN, KCI), which is consistent with the known hardness of CI testing [Shah and Peters, 2020, Polo et al., 2023]. Our proposed Power demonstrates effectiveness, showing slightly higher test power compared to Median when $d_Z = 10$ and 30, though both methods nearly fail when $d_Z$ increases to 50. The above results highlight the intrinsic challenges and distinctive nature of CI testing. Unlike unconditional independence testing and other classical hypothesis testing tasks—for which permutation tests can provide correct calibration under the null—CI testing is fundamentally more difficult. Due to the estimation errors involved in conditioning on potentially high-dimensional variables, no existing CI test can be guaranteed to perform valid hypothesis testing across arbitrary finite-sample scenarios while simultaneously maintaining correct control of the significance level and achieving valid test power.

Table 7: Performance on high-dimensional data.

| Method | $d_Z$ | GCM | RBPT2 | CIRCE | AKE | KNN | CCIT | Median | Power |
|---|---|---|---|---|---|---|---|---|---|
| **Type I Error** | 10 | 0.08 | 0.25 | 0.82 | 0.02 | 0.03 | 0.16 | 0.06 | 0.05 |
| | 30 | 0.03 | 0.28 | 1.00 | 0.06 | 0.05 | 0.15 | 0.03 | 0.02 |
| | 50 | 0.01 | 0.25 | 0.99 | 0.03 | 0.05 | 0.09 | 0.01 | 0.01 |
| **Type II Error** | 10 | 0.50 | 0.68 | 0.02 | 0.62 | 0.90 | 0.58 | 0.56 | 0.47 |
| | 30 | 0.72 | 0.75 | 0.00 | 0.86 | 0.91 | 0.68 | 0.82 | 0.79 |
| | 50 | 0.83 | 0.73 | 0.00 | 0.98 | 0.96 | 0.74 | 0.92 | 0.91 |

## F.4 Comparison with contginuous optimization-based selection

In this section, we further discuss why, unlike in unconditional independence testing, we do not adopt continuous optimization or deep kernels for kernel selection for CI testing. Existing methods for other testing tasks are able to perform kernel parameter selection via continuous optimization, such as in the two-sample test [Gretton et al., 2012b, Kübler et al., 2022] or in unconditional independence testing [Albert et al., 2022, Ren et al., 2024a], and some works have further explored the use of neural networks to parameterize deep kernels [Liu et al., 2020, Xu et al., 2024]. However, the CI testing task differs substantially from these settings, as it not only involves sampling variability but also suffers from model estimation errors. The latter leads to inaccurate gradients of the power-based criterion with respect to the kernel parameters, which undermines the effectiveness of gradient-based optimization. This makes continuous optimization unstable or unable to reliably select effective

kernel parameters, thereby failing to learn optimal kernels or oracle choices for CI testing tasks. To demonstrate this point, we conduct the following experiment using a two-stage optimization framework that performs gradient-based continuous kernel parameter tuning for KCI.

**Setting.** We refer to this two-stage gradient-based method as *Grad*. In the first stage, the Gaussian Process model shares the same configuration as in the previous methods. In the second stage, based on the regression models learned in the first stage, we compute the gradients of the power-based criterion with respect to the parameters of $k_{\mathcal{X}}$ and $k_{\mathcal{Y}}$, and perform parameter updates accordingly. For each iteration of the first-stage regression, one gradient update is performed in the second stage. The ratio-based criterion (i.e., Eq. 9) is inherently more challenging to optimize due to its nonlinearity and potential instability. Following the recommendation of [Wang et al., 2023], we reformulate it as a regularized surrogate objective, which is given by:

$$\check{J} = \hat{C}_{\text{KCIu},\omega}^2 - \lambda \hat{\sigma}_{1,\omega},$$ (55)

where $\lambda$ is the regularization hyper-parameter and $\lambda = 1$.

We conduct experiments on synthetic datasets, where the data is generated following Eq. 48. We set the sample size to $N = 500$, with the data equally split into training and testing sets. In the second stage, we update the kernel parameters of $k_{\mathcal{X}}$ and $k_{\mathcal{Y}}$ using the Adam optimizer with a learning rate of 0.01. We choose total iteration counts of epochs = 10 and epochs = 100 to reflect the behavior of the optimization process at the early stage and after prolonged training, respectively. These two settings are referred to as *Grad-10* and *Grad-100*, and are compared against Median and Power selection strategies. The results are summarized in Table 8, where the last row reports the average test time when $d_Z = 5$.

**Result.** As shown in the results, even with only a small number of updates (*Grad-10*), the performance in terms of Type II error already degrades compared to the Median baseline. As the optimization proceeds further, *Grad-100* still fails to select kernel parameters that lead to improved test power. This suggests that gradient-based methods cannot be directly applied to KCIT for CI testing in the same way as they are used in other settings. We hypothesize that the estimation error inherent to CI testing leads to inaccurate gradients, and the resulting optimization difficulties further cause the procedure to get trapped in poor local minima, preventing it from identifying more effective kernel parameters than those selected by simple heuristics strategy. Moreover, the substantial time cost further renders continuous optimization methods impractical for real-world CI testing applications.

Table 8: Comparison of grid search vs. gradient-based approach for different $d_Z$.

|  | **Method** | $d_Z = 1$ | 3 | 5 | 7 | 9 | **Time (s)** |
|---|---|---|---|---|---|---|---|
| **Type I Error** | Power | 0.04 | 0.03 | 0.03 | 0.03 | 0.06 | 2.267 ± 0.276 |
|  | Median | 0.03 | 0.04 | 0.04 | 0.04 | 0.05 | 1.301 ± 0.186 |
|  | Grad-10 | 0.04 | 0.06 | 0.05 | 0.04 | 0.06 | 11.86 ± 4.136 |
|  | Grad-100 | 0.05 | 0.04 | 0.04 | 0.04 | 0.04 | 119.5 ± 828.6 |
| **Type II Error** | Power | 0.14 | 0.16 | 0.24 | 0.32 | 0.48 | 2.286 ± 0.177 |
|  | Median | 0.22 | 0.25 | 0.32 | 0.52 | 0.58 | 1.244 ± 0.174 |
|  | Grad-10 | 0.22 | 0.28 | 0.34 | 0.49 | 0.68 | 13.44 ± 13.83 |
|  | Grad-100 | 0.27 | 0.31 | 0.46 | 0.61 | 0.73 | 126.5 ± 1019 |

### F.5 Toy experiments.

In this section, we use a top example to demonstrate how the concatenated $\phi(x, z)$ affects the regression learning rate. In vector-valued regression, the learning rate is influenced by the smoothness of the objective function, which is reflected through the parameter $\beta$ (the convergence rate is formally similar to Theorem 7, but with different assumptions). A larger $\beta$ indicates a smoother objective function and is also associated with a faster decay rate of the eigenvalues of the objective function.

**Setting.** We conducted an experiment assuming that the variable $Z \sim \mathcal{N}(0, I_5)$, where $X = Z\mathbf{1}_5$ represents a 5-dimensional vector of ones. We randomly sampled 1000 sets of samples, obtaining the

corresponding kernel matrices $K_x$ and $K_{XZ}$. We performed eigenvalue decomposition on $K_x$ and $K_{XZ}$, and the corresponding eigenvalues $\lambda_i$ were computed.

**Result.** The decay rates of the eigenvalues are shown in Figure 4. The decay rate of the eigenvalues corresponding to $\phi(x, z)$ is significantly slower than that of $\phi(x)$, which corresponds to a slower learning rate and a larger estimation bias under the same sample size.

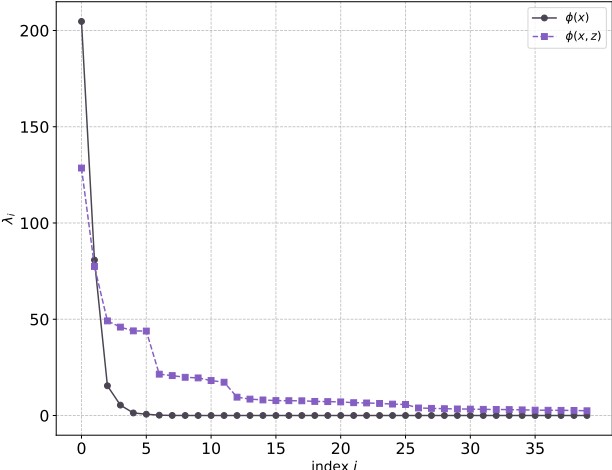

Figure 4: The eigenvalue decay rates for decomposed vs. non-decomposed regression.

