# OpenReview forum: "Practical Kernel Selection for Kernel-based Conditional Independence Test"
_NeurIPS.cc/2025/Conference — NeurIPS 2025 poster_

### Official Review · Reviewer_ik6S · 2025-06-10

**Clarity:** 3
**Significance:** 3
**Originality:** 2
**Rating:** 4
**Confidence:** 4

**Summary:**

This paper proposes a kernel parameter selection method for kernel-based conditional independence (KCI) testing, serving as an alternative to the commonly used median heuristic. The method is theoretically proven to be consistent and explicitly accounts for model estimation bias. Extensive empirical studies on both synthetic and real-world datasets are conducted to validate the effectiveness of the proposed approach.

**Questions:**

Questions. 1. I suggest presenting the results of the ablation study for "On the selectable kernels" and "On the decomposition" separately, so that it is easier to see the impact of each component on the empirical results.

2.The results on the CHILD dataset in Table 2 show that KCIT with the median heuristic for kernel parameter selection performs better. Why isn't it highlighted in bold?

3.I suggest that the authors include the running times of other methods in the main experiments in Section 5.1.3. In addition, I am curious about the running times when these two KCI testing methods are used within the PC algorithm for causal discovery.

4.I'm not sure how the candidate weight list in line 233 is chosen. Is it selected based on the theoretical results provided, or is it chosen empirically?

5.Could you explain in more detail how the p-value of the test is computed in line 237? Is this based on the theoretical results provided in the paper?

6.Why does SelZ not improve the Type II error compared to Median? Please provide an intuitive explanation.

If the authors can address my concerns, I will consider increasing my score.

**Ethical Concerns:**

["NO or VERY MINOR ethics concerns only"]

**Final Justification:**

The authors' rebuttal has addressed my concerns satisfactorily, and I consider this to be a solid paper.

**Limitations:**

yes

**Quality:**

3

**Strengths And Weaknesses:**

Strengths: 1. This paper investigates a valuable topic and provides a well-motivated problem statement.

2. The KCI test is an important tool in machine learning and data analysis, and kernel parameter selection has long been a challenging research problem. This paper proposes a practical parameter selection method that can encourage the broader application of KCI methods in scientific research.

3. The paper is well-structured and easy to follow, the mathematical descriptions are mostly clear and well-written.

4. The paper provide both theoretical proofs and empirical evidence to validate the superiority of the proposed method.

Weaknesses: 1. The comparison of baseline CI testing methods in the simulation study is unfair. To the best of my knowledge, the conditional mean in GCM is typically estimated using kernel ridge regression or XGBoost regression. However, in lines 863 to 864 of the paper, Lasso, a linear regression method, is used instead. This can lead to poor performance of the GCM method under nonlinear simulation settings, which is indeed reflected in its empirical results.

2.The distinction between the methods in the figures is not very clear and requires very careful examination — for example, "Power" and "Median" in Figure 1, and "Median" and "Org" in Figure 2.

3.There is no evidence provided for the parameter selection, such as the choice of the candidate weight list.

---

> ### Author Rebuttal · Authors · 2025-07-30
>
> Dear Reviewer ik6S:
>
> We sincerely thank you for your time and effort in reviewing our submission. Below, we provide our responses to your concerns and questions.
>
> >**W1.** The comparison of baseline CI testing methods in the simulation study is unfair. The model in GCM is typically estimated using kernel ridge regression or XGBoost regression. However, in lines 863 to 864 of the paper, Lasso, a linear regression method, is used instead.
>
> **A.** Thank you for your comment.
> In the following experiments, we modified the original GCM by replacing its regression model with two alternatives:
> (1) KRR using Gaussian processes, with the same parameter initialization and bound settings as in our method;
> (2) XGBoost with the default hyperparameters provided in its official R package, as suggested in the original paper.
> We compared both variants against our Median and Power methods on synthetic datasets, and the results are summarized in the table below.
> We observed that GCM with XGBoost shows substantial improvement in performance: in low-dimensional settings, it effectively controls Type I error while achieving relatively low Type II error.
> However, we also observed that in high-dimensional settings, its Type I error begins to increase. In particular, when $\mathrm{dim}_Z > 7$, our method outperforms GCM-XGBoost.
> We will update the corresponding results and discussion in the next revision of our submission.
>
> | **Type**   | **Model**      | **1**   | **3**   | **5**   | **7**   | **9**   |
> |------------|----------------|---------|---------|---------|---------|---------|
> | **Type I** | GCM-XGBoost    | 0.044   | 0.068   | 0.076   | 0.082   | 0.082   |
> |            | GCM-GPR        | 0.902   | 0.954   | 0.746   | 0.990   | 1.000   |
> |            | Median         | 0.030   | 0.036   | 0.062   | 0.064   | 0.040   |
> |            | Power (Ours)   | 0.048   | 0.062   | 0.054   | 0.056   | 0.036   |
> | **Type II**| GCM-XGBoost    | 0.072   | 0.196   | 0.278   | 0.406   | 0.502   |
> |            | GCM-GPR        | 0.000   | 0.026   | 0.122   | 0.014   | 0.000   |
> |            | Median         | 0.202   | 0.316   | 0.380   | 0.434   | 0.492   |
> |            | Power (Ours)   | 0.142   | 0.246   | 0.314   | 0.326   | 0.418   |
>
>
> >**W2. & Q1.** The distinction between the methods in the figures is not very clear and requires very careful examination.
>
> **A.** Thank you for the suggestion. We will revise the color scheme of the figures and consider splitting Figure 2 into more subplots, which we agree would make it clearer. Thanks again!
>
> >**W3. & Q4.**  I'm not sure how the candidate weight list in line 233 is chosen. Is it selected based on the theoretical results provided, or is it chosen empirically?
>
> **A.** Thank you for your review.
> Our candidate list here is primarily based on empirical intuition, where the parameter space is restricted to a relatively broad but reasonable range.
> We observed that selecting kernel parameters (e.g., bandwidth) that are too large or too small can significantly degrade the performance of CME-based regression, and in some cases even lead to numerical instability.
> This, in turn, adversely affects the final testing power.
> Therefore, we use the median heuristic as a reference point and construct the candidate list by sampling multiple values around it.
> That said, the list can be further adjusted if prior knowledge about the dataset to be tested is available.
>
> >**Q2.** The results on the CHILD dataset in Table 2 show that KCIT with the median heuristic for kernel parameter selection performs better. Why isn't it highlighted in bold?
>
> **A.** Thank you for the careful review. We indeed mistakenly swapped the Power and Median result on CHILD in Table 2. As shown in Table 4 in Appendix F.1.2, Power remains improved over Median on the CHILD benchmark and other datasets.  We will correct it in the revised version, thanks again for pointing it out.
>
>
> >**Q3.** I suggest that the authors include the running times of other methods in the main experiments in Section 5.1.3. In addition, I am curious about the running times when these two KCI testing methods are used within the PC algorithm for causal discovery.
>
> **A.** Thank you for the suggestion.
> We have included the running time of all baseline methods in Table 6 in Appendix F.2.
> Currently, we did not include the full runtime comparison in the main text, as our primary goal was to highlight the trade-off between test power and runtime for the Median and Power.
> We appreciate your comment and will consider moving the runtime comparison of all methods into the main text in the revised version.
> Below, we report the runtime of Median and Power with PC on the SACHs dataset across different sample sizes.
>
> | **Runtime (s)** | **200**         | **400**         | **600**         | **800**         | **1000**        |
> |----------------------|-----------------|-----------------|-----------------|-----------------|-----------------|
> | **Median**           | 22.85 ± 4.25    | 57.70 ± 5.57    | 111.8 ± 13.9    | 270.0 ± 38.3    | 432.2 ± 48.0    |
> | **Power**            | 27.08 ± 6.78    | 74.59 ± 11.0    | 219.6 ± 18.7    | 556.7 ± 49.9    | 958.6 ± 87.7    |
>
>
>
> >**Q5.** Could you explain in more detail how the p-value of the test is computed in line 237? Is this based on the theoretical results provided in the paper?
>
> **A.** In our method, the testing procedure is the same as in the original KCI. After selecting the kernel parameters based on estimated power, we perform testing only once with the final selected parameters. Therefore, the final test can be regarded as a standard test under fixed kernel parameters.
>
> In brief, the KCI testing procedure proceeds as follows: firstly, obtain the residuals with the selected kernel parameters. SVD is then applied to these residuals matrices to extract their principal components (or basis functions), which represent directions in the kernel feature space. According to the theory of partial association, under the null hypothesis $H_0: X \perp Y \mid Z$, the coefficients associated with these basis functions should be uncorrelated, zero-mean, and normally distributed. Consequently, under $H_0$, the KCI statistic approximately follows a weighted sum of zero-mean Gaussian variables, whose null distribution can be approximated by a weighted chi-squared distribution. A detailed derivation is provided in Appendix B.2 of our submission, or in Proposition 5 of the original KCI paper [1].
>
> >**Q6.** Why does SelZ not improve the Type II error compared to Median? Please provide an intuitive explanation.
>
>
> **A.** Thank you for the insightful question. We can answer this result from the definition of conditional cross-covariance operators. From a theoretical standpoint, incorporating $Z$ into the operator input—i.e., using $\phi(X,Z)$ instead of just $\phi(X)$—is necessary for correctly characterizing conditional independence.
> As established in Theorem 8 and Corollary 9 in [2, Appendix A.3], the operator $\Sigma_{Y X|Z} = 0$ only implies a weaker marginal condition ($P_{XY} = \mathbb{E}\_Z[P_{X|Z} \otimes P_{Y|Z}]$), whereas $ \Sigma_{Y(X,Z)|Z} = 0 \iff X \perp Y \mid Z$.
>
> In practice, however, we observe that the choice of kernel parameters for $\phi(Z)$ is not particularly critical for the final testing performance.
> That is, a median heuristic to determine $\phi(Z)$ is often sufficient when using the above operator for general CI testing.
> Furthermore, only in special cases: $X$ and $Y$ are dependent given $Z$ but this dependence varies across different regions of $Z$ in such a way that it cancels out under the weighting induced by the current choice of $\phi(Z)$ can the change of $\phi(Z)$ critically affect the test.
> In these cases, the residuals from kernel regression may appear uncorrelated on average, even though dependence exists locally.
> Changing the choice of $\phi_Z$ (e.g., SelZ) may help to “unmask” this hidden structure by assigning different weights across the $Z$-space.
> But such cancellation cases are rare, and in most scenarios, a reasonable $\phi(Z)$ already suffices.
>
> We hope our response can clarify the confusion and addressed your concerns. We would greatly appreciate it if you could kindly reconsider your score. Should you have any additional questions or suggestions, We’d be happy to clarify them further. Thank you.
>
> **Reference**
>
> [1] Zhang et al. Kernel-based Conditional Independence Test and Application in Causal Discovery, UAI 2011.
>
> [2] Fukumizu K, et al. Dimensionality reduction for supervised learning with reproducing kernel Hilbert spaces. JMLR 2004.

---

> > ### Comment · Reviewer_ik6S · 2025-08-03
> >
> > Dear authors:
> >
> >     Thank you for your clarification — it addressed most of my questions. Additionally, I would like to ask how you selected the parameters when running GCM-KRR?

---

> > > ### Author Response · Authors · 2025-08-03
> > >
> > > Dear Reviewer ik6S,
> > >
> > > Thank you for your careful and thorough review. The GCM-KRR used here is based on the *GaussianProcessRegressor* from the *sklearn* package, with learnable parameters bounded consistently with the settings in our Power and Median experiments. After re-checking the code, we discovered a bug in our implementation of GCM-GPR: when computing residuals, we overlooked the shape mismatch between $X/Y$ and the model predictions, where the former had shape $(N, 1)$ while the latter was $(N,)$. This unintentionally produced a residual matrix of shape $(N, N)$ without raising an error. Thank you for pointing this out and we should have been puzzled by the large performance gap between the XGBoost and KRR models.
> > >
> > > Below are the corrected results of GCM-KRR using Gaussian Processes. From the result, GCM-KRR still shows some inflation in Type I error, but achieves further improvements in test power compared to GCM-XGBoost. Our Power method exhibits better control over Type I error, and when $d_Z > 7$, its Type II error is comparable.
> > >
> > >
> > > | GCM-KRR ($d_Z$) | 1     | 3     | 5     | 7     | 9     |
> > > |-------------------------|-------|-------|-------|-------|-------|
> > > | Type I            | 0.062 | 0.058 | 0.064 | 0.070 | 0.072 |
> > > | Type II           | 0.084 | 0.174 | 0.256 | 0.346 | 0.404 |
> > >
> > >
> > > We appreciate your keen insight, and we will update this part of the results in our revision. Please let us know if you have any further questions or concerns, we’d be happy to discuss them!

---

> > > > ### Comment · Reviewer_ik6S · 2025-08-04
> > > >
> > > > Dear authors,
> > > >
> > > >     Thanks for the response. I will increase my score.

---

> > > > > ### Author Response · Authors · 2025-08-05
> > > > >
> > > > > Dear Reviewer ik6S,
> > > > >
> > > > > We are very grateful for your continued engagement with our work and for taking the time to review our rebuttal. We also appreciate that you pointed out the issues in our additional experimental results and that we were able to address your concerns. Your suggestions will be carefully incorporated into the updated paper to further improve its quality.
> > > > >
> > > > > Best regards,
> > > > >
> > > > > Authors

---

### Official Review · Reviewer_PrW6 · 2025-06-22

**Clarity:** 4
**Significance:** 3
**Originality:** 3
**Rating:** 5
**Confidence:** 2

**Summary:**

The paper presents a new parameter selection method for Kernel Conditional Independencet (KCI) testing. Specifically, the kernel selection criterion is defined as the ratio of the estimates of $C^2_{KCI}$ and $2\sigma_1$, which both depend on the proposed estimator \hat{C}^2_{KCIu}. The convergence rate is provided in Theorem 3 and several experiments are conducted to demonstrate the empirical performance.

**Questions:**

First, I have some minor suggestions on writing/typos:
- Eq.12: I believe it should X/Y \sim f( . ) + E?
- Line 100 - 102: the sentence starts with g(.) but ends with "mappings of (X, Z) and Y onto Z". I believe, (X, Z) --> Y is about h(.) not g(.)?

Second, two questions:
- Line 199, the Boundedness assumption: Could you explain how we can obtain "Then h_{ij} and \hat{h}_{ij} are bounded within [−ν_z ν^2,ν_z ν^2]?
- Of course It's already a lot of experiments and a lot of work for a conference paper, but have you considered checking the empirical performance on other causal structures, i.e., chain or collider?

**Ethical Concerns:**

["NO or VERY MINOR ethics concerns only"]

**Final Justification:**

All my major concerns have been addressed, and I think it is a good contribution to the field.

The paper is well-written, the theoretical results are solid to me, and the experiment results are strong.

**Limitations:**

Yes

**Quality:**

3

**Strengths And Weaknesses:**

Although I cannot understand all the math involved, I like the research question a lot. Such (hyper)-paremeter-choice research questions resolve important practical challenges and potentially have a high impact both for practitioners and for researchers who focus on more applied research, yet are often neglected.

Strength:
- The paper is quite well-written and easy to follow.
- I presume the convergence rate is not bad, i.e., C_2 n^{-1/2} + C_1 n^{-(beta -1) / (2(\beta + p))}, considering the range of \beta and $p$.
- The experiment results are strong: within those methods that can effectively control type-I error, the proposed method "Power" achieves the lowest type-II error in general.

Weakness:
- I would apprecaite a more thorough related work review about kernel (paremeter) selection in the main body.
- It's a bit unclear to me if the U-statistic (eq.5) has been considered by other previous work. I feel the writing a bit unclear here.

---

> ### Author Rebuttal · Authors · 2025-07-30
>
> Dear Reviewer PrW6:
>
> We sincerely thank you for your time, effort, and recognition in reviewing our submission. Below, we provide our responses to your concerns and questions.
>
>
> >**W1.** I would apprecaite a more thorough related work review about kernel (paremeter) selection in the main body.
>
> **A.** Thank you for the feedback. Our discussion of related work is currently placed in Appendix A. We will include a more detailed discussion of existing kernel selection strategies in the main text in the revised version.
>
> >**W2.** It's a bit unclear to me if the U-statistic (eq.5) has been considered by other previous work. I feel the writing a bit unclear here.
>
> **A.** Thank you for sharing your concerns.
> To the best of our knowledge, the form of the U-statistic used here (Eq. 5) is first used in KCI literature.
> It is simple and can be directly derived from the population expectation form of the test statistic.
> KCI itself is an HSIC-like statistic, where the key difference lies in the use of regression residuals instead of raw inputs.
> As we noted in Footnote 1 (under line 165), the U-statistic form of HSIC is more complex due to the requirement of double-centering the kernel matrices.
> In contrast, since KCI operates on residuals, which are already mean-zero by regression, which eliminates the need for explicit centering.
> This leads to a simpler U-statistic form that is not only computationally more efficient, but also more amenable to theoretical analysis.
>
> >**Q1.** Line 199, the Boundedness assumption: Could you explain how we can obtain "Then $h_{ij}$ and $\hat{h}_{ij}$ are bounded within $\[ -{\nu}_z {\nu}^2,{\nu}_z {\nu}^2 \]$ ?
>
> **A.** This bound follows directly from the definition
> $\hat{h}\_{ij} = k_\mathcal{Z}(z_i, z_j) \hat{r}\_{x|z}(s_i, s_j) \hat{r}\_{y|z}(s_i, s_j)$.
> Under the boundedness assumption, we assume that the inner product between any two residuals—both the population value and its empirical estimate—is uniformly bounded by a constant $\nu$.
> This is a standard assumption in the theoretical analysis of kernel ridge regression, often referred to as a moment condition (MOM) [1].
> In our submission, we adopt a simplified form of it.
> In addition, the kernel on $Z$, $k_z(z_i, z_j)$, is assumed to be bounded within $[-\nu_z, \nu_z]$, which is a mild condition satisfied by most commonly used kernels.
> Given these assumptions and the definition of $\hat{h}\_{ij}$, and $h\_{ij}$ corresponds to the expectation form where the CMEs are well estimated, we can directly establish the bound on $\hat{h}\_{ij}$ and $h\_{ij}$.
>
> >**Q2.** Of course It's already a lot of experiments and a lot of work for a conference paper, but have you considered checking the empirical performance on other causal structures, i.e., chain or collider?
>
> **A.** Thank you for your question. Below we provide the experimental results for the chain and collider structures.
> In these experiments, the functional relationships between variables follow the same sampling procedure for $f$ as defined in Eq. (12) of our submission.
> The additive noise is also sampled using the same scheme, with a fixed noise scale of $\gamma = 1$.
> For the collider structure, we define $Z = f_1(X) + f_2(Y) + \epsilon$.
> We set the sample size to 500 and repeat each experiment 500 times.
> All variables are one-dimensional.
> The results are summarized in the table below, where the first two columns correspond to the collider structure and the last two columns to the chain structure.
>
> | **Structure**       | (X     Z ← Y) | (X → Z ← Y) | (Z → X   Y) | (Z → X → Y) |
> |---------------------|---------------|-------------|-------------|-------------|
> | **Type**            | Type I        | Type II     | Type I      | Type II     |
> | **Median**          | 0.04          | 0.49        | 0.03        | 0.16        |
> | **Power**           | 0.03          | 0.37        | 0.02        | 0.13        |
>
>
> **Typos**: (1)  We thank the reviewer for the concerns about the notation in Eq. (12). In this case, our intention was to express a deterministic data generation process where $X$ is constructed from $Z$, rather than to specify a distributional assumption for $X$. For this reason, we believe that using the equality sign is more appropriate in this context.
>
> (2) Thank you for the suggestion. We agree that our original wording here was unclear, and we have revised the paragraph as follows:
> "Theorem 2 can be intuitively understood as asserting that the residuals obtained from regressing any square-integrable functions $g(X, Z)$ and $h(Y)$ onto $Z$ are uncorrelated. Since $g(X, Z)$ can represent any general relationship between $X$ and $Z$, this definition is capable of capturing general CI relationships. However, it requires considering all functions in the $L^2$ space, which is infeasible in practice. "
>
> We hope our response can clarify the confusion and addressed your concerns. Should you have any additional questions or suggestions, We’d be happy to clarify them further. Thank you.
>
> **Reference**
>
> [1] Fischer S, Steinwart I. Sobolev norm learning rates for regularized least-squares algorithms. JMLR, 2020.

---

> > ### Comment · Reviewer_PrW6 · 2025-08-04
> >
> > Thanks a lot for the clarification and additional experiments. It's a nice contribution overall.

---

### Official Review · Reviewer_D99S · 2025-06-30

**Clarity:** 4
**Significance:** 2
**Originality:** 2
**Rating:** 4
**Confidence:** 4

**Summary:**

This paper proposes a simple yet effective kernel selection method for conditional independence testing (CIT). The method enhances the power of CIT by maximizing the ratio of the KCI statistic to its variance. The authors also provide rigorous theoretical guarantees, establishing the convergence rate of their estimator. Empirical results demonstrate that selecting kernel parameters using the proposed approach improves performance by approximately 5%–10% over original methods without kernel selection. Overall, this paper is technically solid and constitutes a valuable contribution to the kernel-based CIT family.

**Questions:**

1. Is it possible to extend your method to scenarios where both X and Y are high-dimensional, despite the well-known fact that kernel methods generally suffer from the curse of dimensionality? Plus, will high-dimensional X, Y obstacle to the proof of theory
2. Given that the topic of this paper is CI **testing**, it would be very beneficial to extend Theorem 3 to bound type I and type II errors of the tests with the tuned kernel.
3. I am not quite clear on how the weight list in line 233 is chosen. Could you provide further explanation or clarification?
4. From Figure 1,3, it can be observed that compared to the ‘Median’ approach, the ‘Power’ method achieves not only a lower type II error (which is the main improvement highlighted in the paper) but also a lower type I error (or at least does not worsen it). This seems somewhat counter-intuitive. This question may be correlated to Q2. Could the authors provide futher explanations for this phenomenon?
5. To better show the improvement of 'Power' beyond 'Median' and other tests, it is recommended to show the densities/histograms of p-values.
6. When developing your Th 3, you used yourTh 7 to handle the estimation error of CME. While Liu et al. developed their Th 6 in [1]. The first term in their error bound exhibits a convergence rate of $n^{-1/3}$. Could you elaborate on what advantages your result offers compared to theirs, in terms of the first term in your (11)?

[1] Liu, Learning Deep Kernels for Non-Parametric Two-Sample Tests, ICML2020.

**Ethical Concerns:**

["NO or VERY MINOR ethics concerns only"]

**Final Justification:**

Most of my concerned have been solved. And the type one error remains to be a challenging open question.

**Limitations:**

yes

**Paper Formatting Concerns:**

Please check Table 2, column CHILD.

**Quality:**

3

**Strengths And Weaknesses:**

Strengths:
1. The method is very simple and intuitive in practice, yet proves to be effective. And the authors provided corresponding theoretical analysis.
2. Cleverly leverage parallelization to carry out the parameter selection, without substantially increasing the overall computational time.
3. Theorem 3 appears to make a significant contribution to the development of the KCI family of methods.

Weakness:
1. My primary concern is that the proposed method can only be applied to improve algorithms within the KCI family, and the improvement may not be substantial.
2. My second concern pertains to the runtime of the algorithm when both the sample size and the dimensionality of Z are high (e.g., dz=30-50, n=1000-2000). The authors have only provided runtime results for cases where dz<10. Given the computational complexity of kernel methods (O(n^3)), I believe this issue could significantly impact the applicability of the proposed method to large-scale datasets.

---

> ### Author Rebuttal · Authors · 2025-07-30
>
> Dear Reviewer D99S:
>
> We sincerely thank you for your time and effort in reviewing our submission. Below, we provide our responses to your concerns and questions.
>
> >**W1.** My primary concern is that the proposed method can only be applied to improve algorithms within the KCI family, and the improvement may not be substantial.
>
> **A.** KCI is a widely used and robust approach for conditional independence testing, as it makes minimal assumptions about the data distribution or underlying functional relationships. Its effectiveness has inspired both methodological advances [1] and real-world scientific applications [2]. In this work, we provide, to the best of our knowledge, the first systematic study of kernel selection for KCI in the context of CIT task. Our method explicitly addresses the model estimation bias inherent in CI testing, leading to improved test power while incurring only moderate additional running time.
>
> >**W2.** My second concern pertains to the runtime of the algorithm when both the sample size and the dimensionality of Z are high. I believe computational complexity could significantly impact the applicability of the proposed method to large-scale datasets.
>
> **A.** Performing CIT in high-dimensional settings remains a highly challenging task.
> As shown in Table 7 in Appendix F.3, not only KCI-based methods, but none of the methods are able to simultaneously control Type I error and achieve sufficient test power under such conditions.
> The primary difficulty lies in the increased complexity of regression in high-dimensional spaces, which affects all regression-based CIT methods similarly.  For permutation-based methods, such setting also causes severe data sparsity during shuffling, limiting their effectiveness.
>
> Regarding the runtime of KCI, a major bottleneck lies in parameter estimation for KRR.
> We have explored faster alternatives such as leave-one-out cross-validation with grid search [3], but found that they often yield poor testing performance due to residual dependence on $Z$.
> Therefore, we adopt Gaussian process (GPR), which, while more computationally intensive, offers better estimation quality.
> Recent advances like GPyTorch [4] provide opportunities for GPR acceleration and can benefit both the Median and Power variants of our method. Exploring the acceleration of KCI is part of our future work.
>
> >**Q1.** Is it possible to extend your method to scenarios where both X/Y are high-dimensional? Plus, will high-dimensional X, Y obstacle to the proof of theory?
>
> **A.** Our method can naturally be extended to cases where both $X$ and $Y$ are multivariate.
> While most existing CIT literature focuses on the setting where $X$ and $Y$ are univariate and the conditioning set $Z$ is multivariate, this is primarily because multivariate $X$ and $Y$ are often treated as collections of multiple univariate regressions.
>
> In KCI-based methods, with using CME as the regressors, increasing the dimensionality of $X$ or $Y$ will potentially increase the complexity of the target function, effectively reducing the smoothness parameter $\beta$.
> This leads to slower convergence rates, as illustrated in Figure 5 in Appendix F.5.
> Nevertheless, the theoretical analysis still holds under the our framework.
> In the following, we evaluate our method with multivariate $X/Y$, extending Eq. 12 by applying an all-one transformation to the original scalar inputs and adding isotropic Gaussian noise.
> We fix the sample size $N = 500$ and $d_Z = 9$, repeat 500 times.
> Empirically, we observe elevated Type I error in high-dimensional settings, which aligns with our analysis.
>
> |               | $d_X, d_Y$| **1**  | **3**  | **5**  | **7**  | **9**|
> |---------------|------------|--------|--------|--------|--------|--------|
> | **Type I**    | Median     | 0.060  | 0.048  | 0.042  | 0.082  | 0.076  |
> |               | Power      | 0.062  | 0.026  | 0.056  | 0.074  | 0.070  |
> | **Type II**   | Median     | 0.512  | 0.070  | 0.092  | 0.082  | 0.078  |
> |               | Power      | 0.446  | 0.066  | 0.056  | 0.064  | 0.074  |
>
>
> >**Q2.** Given that the topic of this paper is CI testing, it would be very beneficial to extend Theorem 3 to bound type I and type II errors of the tests with the tuned kernel.
>
> **A.** In fact, our Theorem 3 does not provide formal guarantees on Type I error control or test power, unlike methods such as GCM [5].
> This limitation arises from fundamental differences in assumptions: GCM relies on a nonlinear additive model, a special case of the general CI setting.
> Such assumption simplifies the null distribution of the test statistic, enabling analytical tractability and formal Type I error and test power guarantees.
> In contrast, KCI targets general CI relationships without assuming a specific functional form, resulting in a more complex and less tractable null distribution.
> This null distribution is typically approximated by a weighted $\chi^2$ distribution, where the weights depend on the eigenvalues of the estimated residual covariance matrix.
> However, the effect of estimation bias on these eigenvalues is intricate and difficult to analyze theoretically.
> As such, establishing precise control over Type I error and test power for KCI remains an open and challenging yet highly valuable problem.
>
> >**Q3.** I am not quite clear on how the weight list in line 233 is chosen. Could you provide further explanation or clarification?
>
> **A.** Our candidate list is primarily guided by empirical intuition, with the parameter space restricted to a broad yet reasonable range.
> We found that kernel parameters that are too large or too small can severely degrade the performance of CME-based regression and even cause numerical instability, which ultimately reduced the test's power.
> To mitigate this, we use the median heuristic as a reference and sample multiple values around it to form the candidate list.
> That said, the list can be further refined if prior knowledge about the target dataset is available.
>
> >**Q4.** From Figure 1,3, ‘Power’ achieves not only a lower type II error but also a lower type I error (or at least does not worsen it). Could the authors provide further explanations for this phenomenon?
>
> **A.** Thank you for your insightful question.
> As we mentioned in our response to Q2, we are currently unable to provide the rigorous theoretical explanation for this phenomenon.
> Intuitively, however, we believe that the choice of $\phi(X)$ should take into account both the dimensionality and the complexity of its dependence on the conditioning set $Z$.
> This aspect is not adequately captured by the Median-based choice.
> And its fixed kernel choice may become increasingly misaligned as $d_Z$ grows, leading to degraded performance.
> In contrast, Power allows flexible kernel parameter selection, enabling better adaptation under both $H_0$ and $H_1$ and thus improving testing performance.
> We conjecture that the remaining deviation from the significance level, particularly when $d_Z$ is large, results from residual bias in estimating the Type I error.
> With the bias improving, we think Power would likely exhibit even greater test power than currently observed.
>
> >**Q5.** It is recommended to show the densities/histograms of p-values.
>
> **A.**  Thank you for your valuable feedback. We will include them in future version.
>
> >**Q6.** While Liu et al. developed their error bound in [6] with exhibiting a convergence rate of
> $n^{-1/3}$. Could you elaborate on what advantages your result offers compared to theirs, in terms of the first term in your (11)?
>
> **A.** The first term in Eq.11 arises from the estimation error of the CME, which is specific to the CIT setting. In contrast, the task in [6] is for two-sample test task and does not involve this term.
> We respectfully assume that your question refers to the second term in our bound, which has a convergence rate of $O(n^{-1/2})$, and may appear faster than the rate reported in [6].
> This is because [6] establishes a uniform convergence result under the additional assumption that the kernel parameterizations are Lipschitz continuous, whereas our result provides only pointwise convergence for fixed kernel parameters.
> We chose not to pursue uniform convergence, as it would require assuming that the learned CME residuals are Lipschitz with respect to the kernel parameters, an assumption that is rather strong and difficult to justify in practice.
> In particular, CME estimation involves matrix inversion operations, which are highly non-smooth and may violate global Lipschitz continuity.
> Moreover, since our method uses grid search rather than gradient-based optimization, pointwise convergence is sufficient for comparing relative test power, which is the central focus of our work.
>
> >**Typos** Check Table 2, column CHILD.
>
> **A.** Thank you for the careful review. We indeed mistakenly swapped the Power and Median result on CHILD, and we will fix it in the revised version.
>
> We hope our response can clarify the confusion and addressed your concerns. We would greatly appreciate it if you could kindly reconsider your score. Should you have any additional questions or suggestions, We’d be happy to clarify them further. Thank you.
>
> **Reference**
>
> [1] Strobl et al. Approximate kernel-based conditional independence tests for fast non-parametric causal discovery. Journal of Causal Inference, 2019.
>
> [2] Kameneva et al. Single-cell transcriptomics of human embryos identifies multiple sympathoblast lineages with potential implications for neuroblastoma origin. Nature Genetics, 2021
>
> [3] Pogodin et al. Efficient Conditionally Invariant Representation Learning, ICLR 2023
>
> [4] Gardner et al. GPyTorch: Blackbox Matrix-Matrix Gaussian Process Inference with GPU Acceleration, NeurIPS 2018
>
> [5] Shah et al. The hardness of conditional independence testing and the generalised covariance measure. The Annals of Statistics, 2020
>
> [6] Feng Liu et al. Learning deep kernels for non-parametric two-sample tests. ICML 2020

---

> > ### Comment · Reviewer_D99S · 2025-08-03
> >
> > To authors:
> >
> > Thanks for the rebuttal. Although the bound on Type I error remains an open problem, authors' rebuttal and extra analyses show the great versatility, computational efficiency, and theoretical property of the proposed method. And I would like to keep the current rating.

---

> > > ### Author Response · Authors · 2025-08-05
> > >
> > > Dear Reviewer D99S,
> > >
> > > We are very grateful for your continued engagement with our work and for taking the time to review our rebuttal. We will continue to explore the conditions for Type I error control of KCI as you suggested, which is indeed a very valuable direction. We sincerely thank you for your constructive feedback. Your suggestions will be carefully incorporated into the updated paper to further improve its quality.
> > >
> > > Best regards,
> > >
> > > Authors

---

### Official Review · Reviewer_USgq · 2025-07-02

**Clarity:** 3
**Significance:** 3
**Originality:** 2
**Rating:** 4
**Confidence:** 4

**Summary:**

This paper addresses the problem of kernel selection in conditional independence testing. It introduces a test power–based criterion along with its corresponding finite-sample estimate. Based on this criterion, a grid search is performed to identify the optimal bandwidth. Experiments on both synthetic and real-world datasets demonstrate that the proposed kernel selection method improves test power compared to the original approach.

**Questions:**

1.what is "K" in line 216?

**Ethical Concerns:**

["NO or VERY MINOR ethics concerns only"]

**Final Justification:**

The authors' rebuttal has addressed my concerns, and I maintain my positive score.

**Limitations:**

yes

**Paper Formatting Concerns:**

No obvious issue

**Quality:**

3

**Strengths And Weaknesses:**

Strengths：
1. The paper provides solid theoretical support for the proposed methodology, and I did not identify any major issues regarding its correctness.
2. The approach is straightforward and enhances efficiency by parallelizing the grid search process, making it practical for real-world applications.

Weakness:
1. While the focus on CI testing is appreciated, the underlying idea of optimizing bandwidth through power maximization has been previously considered in related contexts [1], which somewhat limits the novelty of the contribution.

[1] Feng Liu, Wenkai Xu, Jie Lu, Guangquan Zhang, Arthur Gretton, and Danica J Sutherland. Learning deep kernels for non-parametric two-sample tests. In International conference on machine learning, pages 6316–6326. PMLR, 2020.

2. In the “Overall Procedure”, ω_z  is the median heuristic value (no selection); Fixing ω_z without tuning may limit the flexibility of the kernel ridge regression step and reduce the adaptability of the method in different settings.

3. The experiments are conducted solely using Gaussian kernels. While the proposed method works well under this setting, its applicability to more complex or adaptive kernel designs (e.g., deep kernels) appears limited.

---

> ### Author Rebuttal · Authors · 2025-07-30
>
> Dear Reviewer USgq:
>
> We sincerely thank you for your time and effort in reviewing our submission. Below, we provide our responses to your concerns and questions.
>
> >**W1.**  While the focus on CI testing is appreciated, the underlying idea of optimizing bandwidth through power maximization has been previously considered in related contexts [1], which somewhat limits the novelty of the contribution.
>
> **A.**    Thank you for your comment and for recognizing our focus on kernel selection in CI testing.
> We agree that selecting parameters by maximizing estimated power has been explored in other testing settings, such as two-sample testing and unconditional independence testing.
> However, to the best of our knowledge, this idea has not been effectively extended to CI testing.
> The main challenge lies in the uniqueness of CI testing, which typically requires residual estimation via regression models.
> This additional model estimation step introduces non-negligible model estimation bias, a challenge not present in other testing tasks.
> To address this issue, we propose several key modifications:
>
> (1) Decomposed variant of KCI: We adopt a decomposed form of KCI statistic.
> As demonstrated in Appendix F.5, we find that the original KCI formulation significantly reduces the smoothness of the target function, which in turn slows down the convergence rate of the CME regression.
> The resulting estimation bias compromises the reliability of the test statistic, leading to less effective power-based parameter selection, as demonstrated by the weaker performance of Org and OrgP in Figure 2(a) and 2(b).
> In contrast, the decomposed KCI variant reduces estimation bias and yields more stable test performance.
>
> (2) Grid search instead of gradient-based optimization:  Instead of using gradient-based parameter tuning methods, including those based on neural networks, we adopt a "classical" grid search strategy.
> We found that gradient-based selection methods are often ineffective, likely due to estimation bias introduced during the regression step. This bias distorts the true update direction and prevents reliable identification of effective kernel choices, as illustrated in Appendix F.4.
> Therefore, we adopt a parallelized grid search strategy that is simple, robust, and effective in practice, which serves as an approach well-suited to the unique challenges posed by CI testing.
>
> In summary, while our method is motivated by the principle of test power maximization, it is specifically tailored to the CI testing problem.
> Furthermore, to the best of our knowledge, we are the first to provide convergence rate results for KCI.
> This analysis explicitly accounts for model estimation bias, whereas prior work typically considers only finite-sample randomness.
>
> >**W2.** In the “Overall Procedure”, $\omega_z$ is the median heuristic value (no selection); Fixing $\omega_z$ without tuning may limit the flexibility of the kernel ridge regression step and reduce the adaptability of the method in different settings.
>
> **A.**  We appreciate the reviewer’s comment.
> There appears to be a misunderstanding. The parameter $\omega_z$ mentioned here refers to the kernel parameters associated with $\phi(Z)$ in the decomposed statistic $\phi(X, Z) = \phi(X) \otimes \phi(Z)$.
> That is, it corresponds to the kernel function $k_{\mathcal{Z}}(\cdot, \cdot)$ defined in Eq.5 of the submission, which is not the kernel parameters involved in kernel ridge regression.
> The bandwidth in KRR along with other parameters such the amplitude and the invertible regularization term $\epsilon$ are all learnable.
> We agree that this distinction was not clearly stated here and will clarify it in the revision.
>
> >**W3.** The experiments are conducted solely using Gaussian kernels. While the proposed method works well under this setting, its applicability to more complex or adaptive kernel designs (e.g., deep kernels) appears limited.
>
> **A.** We thank the reviewer for the suggestion regarding kernel choices. In Section 5.1.2 ("On the kernel types"), we compare the use of Gaussian kernels, Laplace kernels, and their combinations to demonstrate that our method can be extended to different kernel families.
> According to the definition of the cross-covariance operator (Theorem 3 in [2]), both $k_{\mathcal{X}}$ and $k_{\mathcal{Y}}$ are required to be characteristic kernels. Among the most widely used and well-understood characteristic kernels are the Gaussian and Laplace kernels, which we have already evaluated in our experiments.
>
> While deep kernels are also characteristic, they typically involve a large number of learnable parameters and rely on gradient-based optimization.
> However, as discussed in our response to W1, we observed that gradient-based methods struggle to reliably optimize even simple kernel parameters, such as the bandwidth of a Gaussian kernel.
> This suggests that extending our method to deep kernels, which require optimizing a high-dimensional parameter space, is currently not feasible due to instability and inaccuracy in gradient estimation.
> We therefore chose to focus on nonparametric kernel families where parameter selection can be performed reliably via grid search.
> Extending to deep kernels is an interesting direction, but it poses substantial challenges that we leave for future work.
>
> >**Q1.** What is "K" in line 216?
>
> **A.** Thank you for your careful review. The constant here $K > 0$ depends on the constants in the assumptions as well as the smoothness parameter $\beta$ of the target function, but it is independent of the sample size $n$ and the confidence level $\delta$.  We apologize for not providing an explanation here, and we will add a clarification in the revised version.
>
> We hope our response can clarify the confusion and addressed your concerns. We would greatly appreciate it if you could kindly reconsider your score. Should you have any additional questions or suggestions, We’d be happy to clarify them further. Thank you.
>
> **Reference**
>
> [1] Feng Liu, Wenkai Xu, Jie Lu, Guangquan Zhang, Arthur Gretton, and Danica J Sutherland. Learning deep kernels for non-parametric two-sample tests. ICML 2020.
>
> [2] Fukumizu K, Gretton A, Sun X, et al. Kernel measures of conditional dependence, NeurIPS 2007.

---

> > ### Comment · Reviewer_USgq · 2025-08-03
> >
> > Thanks for the rebuttal. While the framework is theoretically applicable to more flexible kernels, and gradient-based optimization can indeed be sensitive to estimation noise, the grid search strategy also has its limitations in scalability and expressiveness. Given these trade-offs, I find the authors' approach and clarification reasonable, and I would like to keep my current rating.

---

> > > ### Author Response · Authors · 2025-08-05
> > >
> > > Dear Reviewer USgq,
> > >
> > > We are very grateful for your continued engagement with our work and for taking the time to review our rebuttal. While we acknowledge that the grid search strategy has limitations in terms of scalability and expressiveness, it may currently be the practical choice for the CIT task. We will continue to explore this direction as you suggested. And we sincerely thank you for your constructive feedback. Your suggestions will be carefully incorporated into the updated paper to further improve its quality.
> > >
> > > Best regards,
> > >
> > > Authors

---

### Decision · Program_Chairs · 2025-09-17

**Decision:**

Accept (poster)

**Comment:**

This paper introduces a kernel parameter selection method for the Kernel-based Conditional Independence (KCI) test. Instead of relying on the median heuristic, it decomposes the KCI statistic to isolate the kernel applied on the conditioning set, treating its parameters as trainable. These parameters are optimized by maximizing the ratio of the estimated test statistic to its variance, a proxy for test power at large sample sizes. The resulting kernel choice enhances conditional independence test power with minimal added computational cost. Theoretically grounded consistency guarantees and empirical improvements, approximately a 5–10% gain in power on synthetic and real datasets, demonstrate its strong practical value.

In a kernel-based testing framework, kernel selection is an important issue. The proposed approach is interesting, and all the reviewers agreed on accepting the paper. Thus, I also vote for acceptance.